# Monitoring the seasonal dynamics of soil salinization in the Yellow River Delta of China using Landsat data

Hongyan Chen, Gengxing Zhao[*], Yuhuan Li, Danyang Wang, Ying Ma

*National Engineering Laboratory for Efficient Utilization of Soil and Fertilizer Resources, College of Resources and Environment, Shandong Agricultural University, Taian 271018, China*

[*]Corresponding author: Gengxing Zhao, College of Resources and Environment, Shandong Agricultural University, No. 61 Daizong Street, Tai'an, Shandong 271018, PR China. E-mail: zhaogx@sdau.edu.cn

**Abstract.** In regions with distinct seasons, soil salinity usually varies greatly by season. Thus, the seasonal dynamics of soil salinization must be monitored to prevent and control soil salinity hazards and to reduce ecological risk. This article took the Kenli District in the Yellow River Delta (YRD) of China as the experimental area. Based on Landsat data from spring and autumn, improved vegetation indices (IVIs) were created and then applied to inversion modeling of the soil salinity content (SSC) by employing stepwise multiple linear regression, back propagation neural network and support vector machine methods. Finally, the optimal SSC model in each season was extracted, and the spatial distributions and seasonal dynamics of SSC within a year were analyzed. The results indicated that the SSC varied by season in the YRD, and the support vector machine method offered the best SSC inversion models for the precision of the calibration set ($R^2$>0.72, RMSE <6.34 g/kg) and the validation set ($R^2$>0.71, RMSE<6.00 g/kg, and RPD>1.66). The best SSC inversion model for spring could be applied to the SSC inversion in winter ($R^2$ of 0.66), and the best model for autumn could be applied to the SSC inversion in summer ($R^2$ of 0.65). The SSC exhibited a gradual increasing trend from the southwest to northeast in the Kenli District. The SSC also underwent the following seasonal dynamics: soil salinity accumulated in spring, decreased in summer, increased in autumn, and reached its peak at the end of winter. This work provides data support for the control of soil salinity hazards and utilization of saline-alkali soil in the YRD.

**Keywords:** Soil salinity; Remote sensing; Vegetation index; Multispectral imaging; Seasonal dynamics

## 1. Introduction

Saline soils are widespread throughout the world, especially in arid, semiarid and some subhumid regions, and they cause severe environmental degradation that can impede crop growth and overall regional production (Metternicht and Zinck 2003). Moreover, as a form of land degradation and an ecological environment hazard, soil salinization can degrade soil quality and lead to ecosystem risks (Huang et al. 2015; Zhao et al.

2018). Therefore, the scientific treatment and utilization of saline soil are of great significance to regional agricultural production and ecological risk reduction. Thus, the degree, geographical distribution and dynamics of soil salinization must be determined in real time for the prevention and control of soil salinity hazards and the management and utilization of saline soil (Melendez-Pastor et al. 2012; Yang et al. 2018).

Remote sensing technology provides an important and rapid approach for the quantitative monitoring and mapping of soil salinization (Dehni and Lounis 2012; Tayebi et al. 2013; Shoshany et al. 2013; Sidike et al. 2014; Wu et al. 2014; Guo et al. 2015; Sturari et al. 2017). Multispectral satellite data, such as Landsat, SPOT, IKONOS,

QuickBird and the Indian Remote Sensing (IRS) series of satellites, have often been used to map and monitor soil salinity and other properties due to the low cost and the ability to map extreme surface expressions of salinity (Dwivedi et al. 2008; Abbas et al. 2013; Allbed et al. 2014; Mahyou et al. 2016; Mehrjardi et al. 2008; Yu et al. 2010; Ahmed and Iqbal 2014; Rahmati and Hamzehpour 2016). Extensive studies have shown

that models based on multispectral satellite data are still the preferred soil salinity

mapping method over large spatial domains (Allbed and Kumar 2013; Scudiero et al.

2015; Taghizadeh-Mehrjardi et al. 2014).

To a certain extent, information on the damage to vegetation caused by soil

salinization can help identify the degree and trend of soil salinization. Therefore,

traditional vegetation indices (VIs), such as the normalized difference vegetation index

(NDVI), ratio vegetation index (RVI) and difference vegetation index (DVI), can be

used as indicators to determine the degree of soil salinization (Elmetwalli et al. 2012; Li

et al. 2013; Goto et al. 2015). However, the accuracy of the models based on traditional

VIs must be improved (Iqbal 2011). Traditional VIs involve data from only the bands in

the visible and near-infrared regions and do not consider other bands (such as shortwave

infrared band, which has a large amount of information), and performing

comprehensive analyses with these indices is difficult. Significant correlations often

occur between traditional VIs that only take advantage of the data from two bands,

which can distort the model results (USGS, 2013). Therefore, whether the addition of

data from the shortwave infrared band which has long wavelengths and contains

considerable information, can improve the accuracy and stability of inversion models of

the soil salinity content (SSC) warrants further study.

Existing studies have primarily focused on SSC inversion models for a single study

area at a specific time (Herrero and Castañeda 2015; He et al. 2014). Nevertheless, in

regions with distinct seasons, applying the same inversion model to quantitatively

analyze the SSC in different seasons is not adequate. For one thing, the changes in soil

moisture are obvious due to the great differences in rainfall and evaporation between

different seasons, thus, because being closely related to soil moisture, soil salinity varies usually greatly by season. Moreover, considerable differences occur in the coverage and growth of vegetation between different seasons, which have a great influence on VIs. Therefore, seasonal SSC inversion models are necessary to improve the accuracy of

SSC modeling and enhance our ability to monitor regional soil salinization continuously and in real time.

The Yellow River Delta (YRD) is located at the junction of the Beijing-Tianjin-Hebei metropolitan area and Shandong Peninsula, and it lies within the efficient

ecological economic zone of China, which has obvious geographical advantages. In this region, the land resources are rich with nearly 550,000 ha of unused land, but soil salinization is a widespread and serious concern (Mao et al. 2014). Approximately 85.7% of the region is covered by saline soil, and the amount of coastal saline soil has exhibited an increasing trend in recent years. As the main risk to farmland ecosystems

in this region, soil salinization can result in large reductions in agricultural and fragile ecological environments, which may influence the development of the regional economy and society (Yang et al. 2015; Weng et al. 2010). Therefore, it is particularly necessary to monitor the seasonal dynamics of soil salinization in this region.

The objectives of this paper were (1) to build optimal SSC inversion models for different seasons according to the soil salinity conditions and (2) to map the spatial distribution and seasonal dynamics of SSC in the YRD of China. Specifically, VIs were generated by introducing data from the shortwave infrared band (SWIR) of Landsat

data. The SSC inversion models in spring and autumn were built using stepwise multiple linear regression (SMLR), back propagation neural network (BPNN) and support vector machine (SVM) methods, and the best models for spring and autumn were selected and applied to the other seasons. Once the optimal soil salinity

inversion model was determined for each season, it was then applied to map the SSC distribution and analyze the seasonal SSC dynamics.

## 2. Materials and methods

### 2.1 Study area

The study area was the Kenli District in the YRD region (37°24′~38°06′N, 118°14′~119°11′E) located in Dongying City, Shandong Province, China, and on the southern shore of the Bohai Sea (Figure 1). This area has a characteristic plain landscape and coastal saline soil type, and the following three types of soil subgroups are present in this area: tidal soil, salinized tidal soil and coastal tidal saline soil. The

soil parent material is Yellow River alluvial material, and the soil texture is light. The salt in groundwater can easily reach the soil surface with the evaporation of water from the soil. Thus, salt accumulates on the soil surface, while it is relatively rare in the middle and lower parts of the soil profile (below the core soil). The main types of land use in this area are cultivated land, unused land and grassland. The main crops are

wheat, corn, rice and cotton. The main natural vegetation includes white grass, reed, horse trip grass, tamarix and suaeda. Because of the low and flat terrain, high groundwater table, high mineralization rate, poor drainage conditions and the infiltration and mounting of seawater associated with the Yellow River in this region, soil salinization at the surface is generally severe and widespread, and the associated

ecological risk is profound (Yang et al. 2015; Weng et al. 2010). Due to the temperate climate and the occurrence of four distinct seasons, the soil salt content exhibits seasonal dynamics. The soil salinization process in the region is shown in Figure 2.

*2.2 Soil sampling and chemical analyses*

To achieve an accurate representation of the seasonality, we selected April, August, November and February (in the following year) to represent the spring, summer, autumn, and winter seasons, respectively. According to the climate characteristics and soil salinization conditions in the different seasons, the samples collected in spring and

autumn were used to develop the SSC inversion models, while the samples from winter and summer were used to validate the inversion models. The following soil samples were collected: 92 spring samples were collected from April 27-May 2, 2013; 30 summer samples were collected from August 14-15, 2013; 110 autumn samples were collected from November 9-13, 2013; and 56 winter samples were collected from

February 26-29, 2014. Sample points were designated by considering the degree of soil salinization, morphology and microtopography of the soil surface, and uniformity of the sample distribution (Figure 1). Topsoil samples were collected at each sample point at a depth <20 cm, and GPS coordinates were recorded. *In situ* environmental information was also recorded. The collected soil samples were naturally air dried, crushed, purified,

passed through a 2 mm sieve and mixed evenly. The concentrations of $Cl^-$, $SO_4^{2-}$, $CO_3^{2-}$, $HCO_3^-$, $K^+$, $Na^+$, $Ca^{2+}$ and $Mg^{2+}$ in extracted solutions of a 1:5 soil-water mixture were measured. The SSC was defined as the combined concentration of the eight ions mentioned above.

## 2.3 Acquisition and pretreatment of imaging data

Multispectral Landsat data were acquired in line with the sample collection time. We employed Landsat 7 ETM+ data from May 6, 2013 and Landsat 8 OLI data from August 18, 2013, November 6, 2013, and February 26, 2014. The Landsat 7 ETM+ data included the following bands: one panchromatic band (520–900 nm); four multispectral bands in the visible and near-infrared wavelength range (blue, 450–515 nm; green, 525–605 nm; red, 630–690 nm; and NIR, 775–900 nm); and two shortwave infrared (SWIR) bands (1550–1750 nm and 2090–2350 nm). The Landsat 8 OLI data had the same bands as ETM+, but the band ranges were slightly different. Image pretreatment, including geometric rectification, radiation calibration and atmospheric correction, was conducted in ENVI 5.1 software from Exelis Visual Information Solutions. Geometric rectification was completed in reference to the 1:10000 terrain map of the study area. The radiation calibration and Fast Line-of-Sight Atmospheric Analysis of Spectral Hypercubes (FLAASH) atmospheric correction were subsequently applied. The output images were projected to the Gauss–Kruger coordinate system and cropped to the study area. The water body, building and traffic land areas were then masked according to the current land use situation. Finally, the reflectance of the samples was extracted from the processed images using ArcGIS 10.1 software.

## 2.4 Calculation and improvement of vegetation indices

The extended vegetation indices (EVIs) were all calculated based on Landsat data by adding the SWIR band data to the traditional VIs. These EVIs included the extended normalized difference vegetation index (ENDVI, (NIR+SWIR-R)/(NIR+SWIR+R)), extended difference vegetation index (EDVI, NIR+SWIR-R) and extended ratio vegetation index (ERVI, (NIR+SWIR)/R). The SWIR band refers to either of the two

SWIR bands in Landsat data. The correlations between the SSC and EVIs were analyzed, and the EVIs with significant correlation coefficients were selected as the improved vegetation indices (IVIs). Finally, the IVIs were used as inputs to the SSC inversion models.

### 2.5 Inversion model construction and optimization

First, the soil samples collected in spring and autumn were sorted and separated according to the SSC. Two-thirds of the samples were selected for the calibration set, and the remaining samples were used as the validation set. Of the 92 samples collected

in spring, 62 were used for calibration, and the other 30 were used for validation. Of the 110 samples collected in autumn, 74 were used for calibration, and the other 36 were used for validation. Second, the SSC inversion models for spring were built by employing the SMLR, BPNN and SVM methods based on the VIs and corresponding IVIs. The performance of the SSC inversion models was evaluated by the coefficient of

determination ($R^2$), root-mean-square error (RMSE) and ratio of performance to deviation (RPD). Using the same procedures, the SSC models for autumn were built using the IVIs, and the best model was selected. Finally, the best models for spring and autumn were selected and applied to the summer and winter data, and the optimal SSC inversion models according to the soil salinization conditions in different seasons were

then selected.

For the SMLR method, the variance inflation factor (VIF) was set to less than 5 to control for multicollinearity. The BPNN method was conducted using the MATLAB R2012a program. During the calculation, the transfer functions of the hidden layer and

the output layer were set to tansig and logsig, respectively. The network training

function was traingdx, and the learning rate, maximum training time and model

expectation error were set to 0.01, 15000 and 0.01, respectively. The SVM models were

built in the Libsvm 3.11 toolbox in MATLAB R2012a, and the 4th SVM type (v-SVR)

and 2nd kernel function (RBF) were selected. The penalty parameter C and the kernel

parameter g of the RBF were determined according to the minimum mean-squared

deviation by using the cross-validation and grid search methods.

### *2.6 SSC distribution mapping and year-round dynamics analysis*

The reflectance spectra were extracted from the Landsat data of four seasons in the

study area, and the seasonal IVIs were calculated. The SSC distribution maps of four

seasons were then obtained via calculations based on the corresponding optimal models.

The spatial distribution characteristics and seasonal dynamics of soil salinity in the

YRD were analyzed and compared.

     The methodological flow of this article is shown in Figure 3.

### 3. Results

### *3.1 Soil sample data*

The statistical results of the SSC samples from the four seasons (the upper half of Table

1) showed that the SSC in the study area remained high with a mean >5.32 g/kg

throughout the year. As determined from the minimum, maximum and mean values, the

SSC reached its maximum concentration in winter (mean=9.50 g/kg) and varied by

season. Because the coefficients of variation for all four seasons were greater than 1.00, the overall SSC gradient was obvious, especially in winter and spring.

### 3.2 Improved vegetation indices (IVIs)

In spring, the correlation coefficients between the EVIs and the SSC of the soil samples were -0.52 for ENDVI, -0.69 for ERVI and -0.70 for EDVI. Similarly, in autumn, the correlation coefficients between the EVIs and the SSC of the soil samples were -0.73 for ENDVI, -0.69 for ERVI and -0.69 for EDVI.

The results showed that the correlation coefficients between the ERVI or EDVI and SSC were significant ($R^2$>0.69; P<0.01) in spring. Based on these findings, ERVI and EDVI were selected as the IVIs for spring, while ENDVI and ERVI were selected as the IVIs for autumn. For each season, the chosen IVIs and their corresponding VIs were used to build the SSC inversion models.

### 3.3 Best SSC inversion models and their application to different seasons in the YRD region

#### 3.3.1 SSC inversion models with VIs and IVIs

The results of the SSC inversion models in spring based on IVIs are shown in
Table 2. The performances of the three modeling methods were compared, which indicated that the SVM models had the highest prediction accuracy, followed by the BPNN models, and the SMLR models had the lowest accuracy. In terms of the

calibration values, the SVM models based on the IVIs had the best and most stable SSC inversion accuracies for both the calibration set ($R^2$>0.72, RMSE<6.34 g/kg) and the validation set ($R^2$>0.71, RMSE<6.00 g/kg, and RPD>1.66). These models were then selected as the best SSC inversion models for the SSC in spring and autumn.

The calibration and validation precision of the SSC inversion models in spring and autumn are shown in Figure 4.

*3.3.2 Application of the best SSC inversion models with IVIs in different seasons*

10    The best *SSC* inversion models for spring and autumn were applied to estimate the SSC in summer and winter, respectively. Based on the estimation accuracy (Table 3), the best SSC inversion model for spring could be applied to estimate the SSC in winter with $R^2$ of 0.66 and RMSE of 7.57 g/kg. Meanwhile, the best SSC inversion model for autumn could also be applied to estimate the SSC in summer, resulting in $R^2$ of 0.65 and

15    RMSE of 3.60 g/kg. In response to the soil salinity conditions, the SSC inversion model for spring based on the IVIs in combination with the SVM method was selected as the optimal SSC model for spring and winter, while the SSC inversion model for autumn based on the IVIs in combination with the SVM method was selected as the optimal SSC model for autumn and summer.

### *3.4 Distribution and seasonal dynamics of SSC in the YRD region*

#### *3.4.1 Distribution of SSC across the four seasons*

Based on the processed Landsat data and the optimal SSC inversion model for each season, SSC inversion maps were obtained for all four seasons. The descriptive statistics of the inversed SSC are shown in the lower half of Table 1, and these values were close to those from the collected samples (the upper half of Table 1). The inversion results also showed that the SSC was highest in winter, followed by the SSC in spring, and the SSCs in autumn and summer were relatively low. Therefore, the ecological risk in winter and spring was high, while the ecological risk in autumn and summer was relatively low.

According to the classification standard of coastal saline soil in the semihumid area of China, the study area was divided into five grades as follows: nonsaline soil, mild saline soil, moderate saline soil, severe saline soil and solonchak. The distributions of the soil salinity grades in the four seasons were mapped (Figure 5) and showed similar characteristics. There was a gradual increasing trend in soil salinity from the southwest to northeast in the study region. The main reason for this gradual increase in SSC is that the terrain in the southwest part of the study area is high and flat and the land is used for agricultural production with relatively less soil salinity hazards, while the central part of the region near the banks of the Yellow River has alternating hillocks, slopes and depressions, which were formed by the repeated diversion of the Yellow River, thus, each grade of soil salinization was also alternately distributed. The northeast part of the region, which has low terrain and is closest to the sea, exhibited the most severe soil salinization and hazards.

*3.4.2 Seasonal dynamics of SSC*

The number and proportion of pixels per SSC grade were calculated for each season (Table 4). Figure 5 and Table 4 demonstrate that the SSC in the study area clearly differed among the four seasons. The SSC in spring consisted primarily of moderate saline soil, severe saline soil and solonchak (combined proportion of 90.05%). In summer, the areas of the four grades from mild saline soil to solonchak were relatively uniform (each grade accounting for 22–28%). The SSC during autumn was largely dominated by severe saline soil and solonchak (combined proportion of 77.75%). In winter, the SSC was principally severely saline and solonchak (combined proportion of 99.19%, of which severe saline soil accounted for 80.71%).

The seasonal SSC inversion values and the proportion of pixels per SSC grade indicated that the change in SSC between different seasons was relatively apparent. The degree of soil salinization was lowest in summer, and the SSC in autumn was relatively low except for the solonchak in coastal areas. In spring, soil salinization became more obvious, as most of the study area belongs to the moderate to severe saline soil and solonchak groups. Meanwhile, soil salinization was the most severe in winter. In summary, soil salinity in the study area usually accumulated in spring, decreased in summer, increased in autumn and reached its peak at the end of winter.

**4. Discussion**

In this work, spatial distribution maps of SSC in the study region in four seasons were obtained. Soil salinity exhibited a gradually increasing trend from the southwest to northeast. Soil salinization at the surface was generally severe and widespread, and soil

salinity hazards were obvious, indicating that the ecological ecosystem was fragile. These conditions can lead to a high ecological risk, especially in winter and spring. The results provide data support for soil ecological risk assessments to promote the prevention and control of soil salinity hazards and improve agricultural production and

the sustainable development of the ecological environment. Moreover, different measures for the treatment and utilization of saline soil should be implemented according to the different regions and seasons.

In this experiment, we introduced the SWIR band and proposed an improved

vegetation index to increase the accuracy of SSC inversion models. The spatial distributions of SSC in the four seasons showed similar characteristics. The soil salinity exhibited a gradual increasing trend from the southwest to northeast in the study region, and this distribution pattern was consistent with the results of other studies (Weng et al. 2010; Yang et al. 2015). Weng et al. (2010) also established a SSC remote sensing

revision model using the data from 2153~2254 nm and 1941~2092 nm in the YRD region and achieved good results with a validation RMSE of 0.986 and $R^2$ of 0.873.

The best SSC inversion models for spring and autumn were based on different IVIs. In spring, the weather is characteristically dry and windy, and strong evaporation occurs.

Moreover, the coverage of natural vegetation is low, but certain crops, such as wheat and corn, are in a vigorous growth stage, which results in strong vegetation reflectance. Generally, the RVI and DVI are sensitive to vegetation, especially when vegetation coverage is high. Thus, the inversion accuracies based on the ERVI and EDVI were higher than those based on the other vegetation indices. In autumn, rainfall and

temperature are reduced, and there is little natural vegetation coverage. Moreover, in autumn, cotton is collected, and only withered cotton leaves and rods remain in the field. As wheat is only beginning to emerge out of the soil in autumn, there is limited crop coverage. Therefore, the reflectance spectra of vegetation are relatively weak in autumn.

NDVI has low sensitivity to high vegetation areas and is thus more suitable for the monitoring of low and moderate vegetation coverage areas. Accordingly, the inversion accuracies based on ENDVI and ERVI were higher than those based on the other vegetation indices. Without considering the influence of some factors (e.g., soil moisture and temperature), the models can be used for the regional remote sensing

inversion of SSC in the study area, but the prediction accuracy still needs to be improved. Therefore, the influence of certain key factors and the uncertainty of the quantization will be further studied in the future to improve the SSC prediction accuracy.

       The seasonal dynamics of SSC are closely related to the climate of the study area.

Under drought, windy weather and strong evaporation conditions in spring from March to May, soil salt aggregates at the soil surface as the soil moisture increases, thereby forming the first peak of salt accumulation. At this time, 90.05% of the area is covered by moderate saline soil, severe saline soil and solonchak. Rainfall and floods occur in the summer from June to August, and as precipitation infiltrates into the soil, the soil

surface is desalinated with uniform proportions of mild saline soil to solonchak. In autumn from September to November, rainfall decreases, and SSC increases slightly. In addition, the area is largely dominated by severe saline soil and solonchak (combined proportion of 77.75%). Due to drought in winter from December to February and combined with decreased evaporation, soil salinization is relatively severe and remains

latent at the soil surface, with 99.19% of the area covered by severe saline and solonchak. By the end of winter, the SSC reaches its peak. The study by Lu et al. (2016) indicated that SSC exhibits seasonal variations in the YRD and that the SSC in spring is higher than that in autumn in the Kenli District, which is consistent with our results.

Based on the time point data, the results indicated that the SSC inversion model for spring could be applied to the SSC inversion in winter, while the SSC inversion model for autumn could be applied to the SSC inversion in summer in the YRD. These model selection results may be due to the short time intervals and the similar soil salt contents

and climatic conditions between February and April as well as between August and November in the YRD. For more accurate responses to the dynamic changes in soil salinity, a period of SSC data should be selected as the seasonal salt data, which will be studied further in the future.

**5. Conclusions**

In this experiment, the ERVI and EDVI were the IVIs for spring, while the ENDVI and ERVI were the IVIs for autumn. These models based on the IVIs and the SVM method were selected as the best SSC inversion models for spring and autumn. The experimental results contribute to the quantitative and accurate monitoring of soil

salinization via multispectral imaging and provide data and technical support for the management and utilization of saline soil and the protection of the ecological environment.

This experiment indicated that the best inversion model for spring could be applied for the SSC inversion in winter. Thus, the optimal SSC model for spring and winter was selected in response to the soil salinity conditions. At the same time, the best inversion model for autumn could also be applied for the SSC inversion in summer, and it was

selected as the optimal SSC model for autumn and summer in the study region.

In the YRD region, the spatial distribution of SSC showed a gradual increasing trend from the southwest to northeast. The seasonal dynamics of SSC indicated that soil salinity accumulated in spring, decreased in summer, increased in autumn and reached

its peak at the end of winter. These results were consistent with the results of field sampling, which showed that the SSC was highest in winter followed by spring and autumn, and the lowest SSC occurred in summer.

Author contributions. Hongyan Chen analyzed the data and prepared the manuscript. Gengxing

Zhao developed the framework for the study. Danyang Wang and Ying Ma collected and analyzed the data. Yuhuan Li provided technical support throughout different stages of the study. All coauthors provided a manuscript review.

Competing interests. The authors declare that they have no conflicts of interest.

**Acknowledgments**

This work was financially supported by the National Natural Science Foundation of China under grant numbers 41877003 and 41671346; the Natural Science Foundation of Shandong Province, China, under grant numbers ZR2019MD039; the National Science and Technology

Support Program of China under grant number 2015BAD23B0202; the Funds of Shandong

"Double Tops" Program under grant number SYL2017XTTD02; and Shandong Province key R

& D Plan of China under grant number 2017CXGC0306.

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

Tables

Table 1. SSC descriptive statistics of samples and inversion

| | Seasons | Minimum (g/kg) | Maximum (g/kg) | Mean (g/kg) | Standard deviation (g/kg) | Coefficient of variation | Sample points |
|---|---|---|---|---|---|---|---|
| Soil samples SSC | Spring | 1.10 | 46.70 | 8.60 | 11.50 | 1.34 | 92 |
| | Summer | 1.34 | 29.20 | 5.32 | 5.95 | 1.12 | 30 |
| | Autumn | 0.90 | 36.70 | 7.80 | 8.20 | 1.05 | 110 |
| | Winter | 2.00 | 61.50 | 9.50 | 12.00 | 1.26 | 56 |
| Inversion SSC | Spring | 0.86 | 53.44 | 8.79 | 10.26 | 1.17 | |
| | Summer | 1.00 | 35.50 | 7.00 | 4.18 | 0.60 | |
| | Autumn | 0.82 | 35.15 | 8.28 | 9.21 | 1.11 | |
| | Winter | 1.12 | 58.10 | 9.21 | 13.78 | 1.50 | |

Table 2. Inversion models of SSC with IVIs from Landsat data

| Modeling methods | Spring | | | | | Autumn | | | | |
|---|---|---|---|---|---|---|---|---|---|---|
| | Calibration set | | Validation set | | | Calibration set | | Validation set | | |
| | $R^2$ | RMSE (g/kg) | $R^2$ | RMSE (g/kg) | $RPD$ | $R^2$ | RMSE (g/kg) | $R^2$ | RMSE (g/kg) | $RPD$ |
| SMLR | 0.42 | 9.03 | 0.62** | 6.83 | 1.36 | 0.65** | 3.42 | 0.56 | 3.81 | 2.01 |
| BPNN | 0.60** | 7.56 | 0.57** | 7.30 | 1.47 | 0.72** | 3.39 | 0.68** | 3.38 | 2.15 |
| SVM | 0.72** | 6.34 | 0.71** | 6.00 | 1.66 | 0.75** | 3.48 | 0.78** | 3.02 | 2.56 |

5    Significance levels: [**] 0.01

Table 3. Application of the best SSC inversion models

| | The best inversion model for spring | | The best inversion model for autumn | |
|---|---|---|---|---|
| | $R^2$ | RMSE (g/kg) | $R^2$ | RMSE (g/kg) |
| Summer samples (30) | 0.23 | 5.31 | 0.65** | 3.60 |
| Winter samples (56) | 0.66** | 7.57 | 0.28 | 10.98 |

Significance levels: [**] 0.01

Table 4. The number and proportion of pixels per SSC grade across four seasons

| Grades | Spring | | Summer | | Autumn | | Winter | |
|---|---|---|---|---|---|---|---|---|
| | Number of pixels | Proportion % | Number of pixels | Proportion % | Number of pixels | Proportion % | Number of pixels | Proportion % |
| Nonsaline soil (<2.0 g/kg) | 10705 | 0.67 | 16 | 0 | 46439 | 2.89 | 3 | 0 |
| Mild saline soil (2.0~4.0 g/kg) | 84805 | 5.29 | 450331 | 28.07 | 127262 | 7.93 | 12 | 0 |
| Moderate saline soil (4.0~6.0 g/kg) | 451291 | 28.13 | 427216 | 26.63 | 182589 | 11.37 | 13045 | 0.81 |
| Severe saline soil (6.0~10.0 g/kg) | 597607 | 37.25 | 371641 | 23.16 | 305762 | 19.05 | 1294867 | 80.71 |
| Solonchak (>10.0 g/kg) | 459989 | 28.67 | 355193 | 22.14 | 942345 | 58.70 | 296470 | 18.48 |

Figures

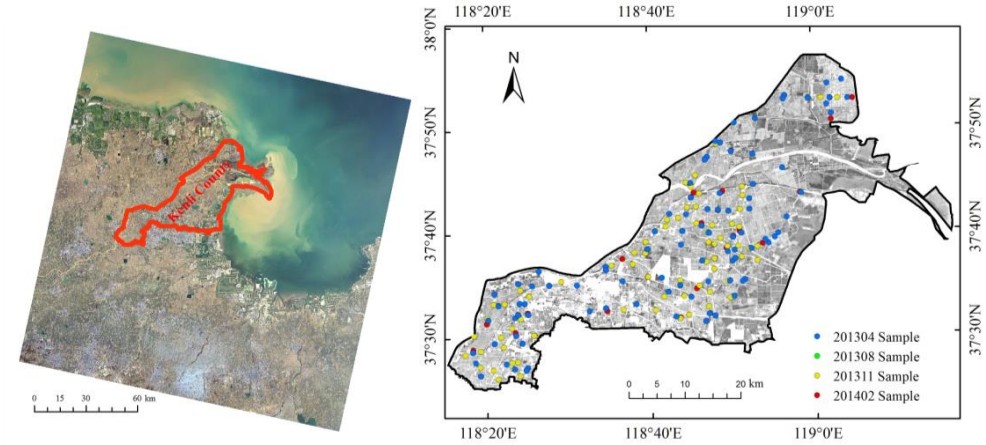

Figure 1. Location of the study area and sampling points

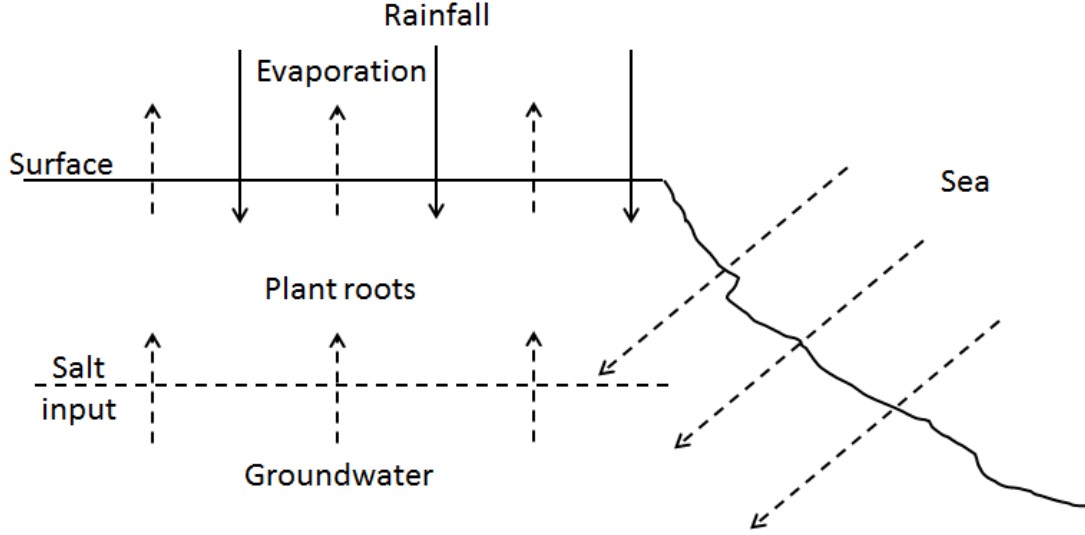

Figure 2. Soil salinization process in the study area

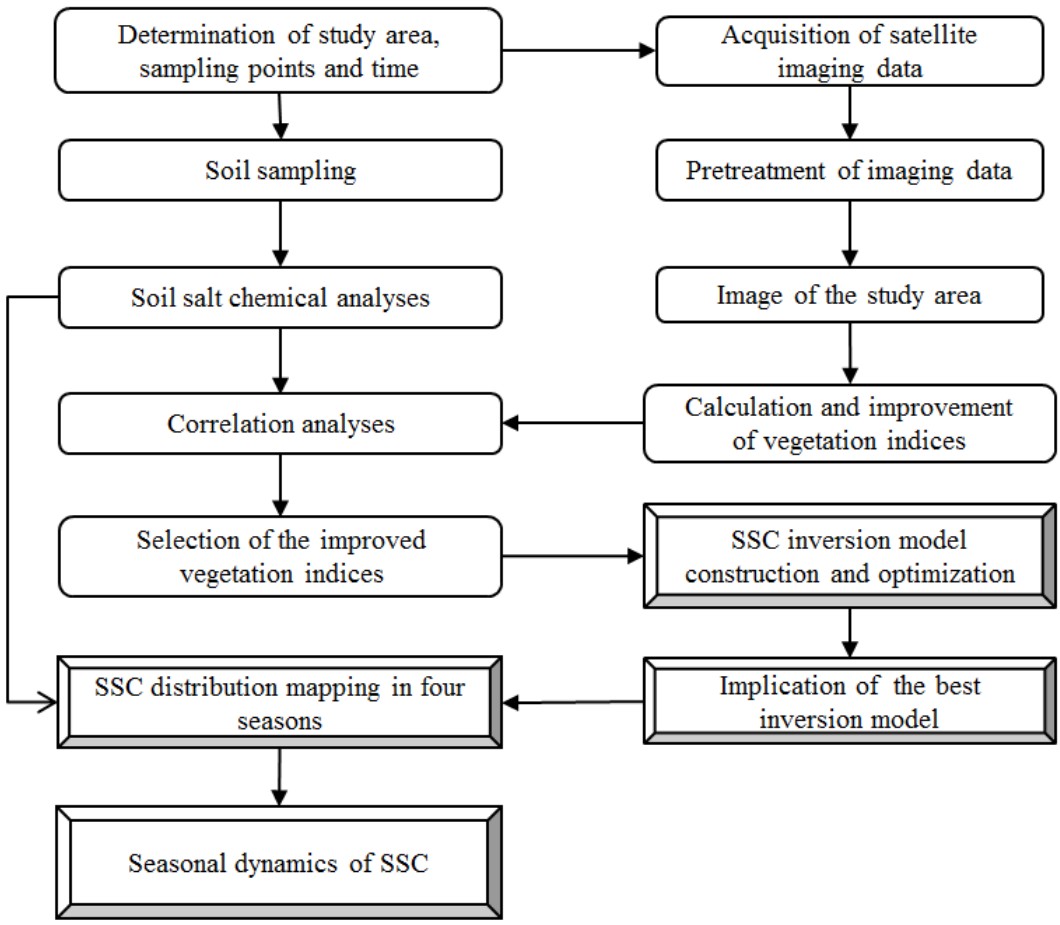

5    Figure 3. Methodological flow chart

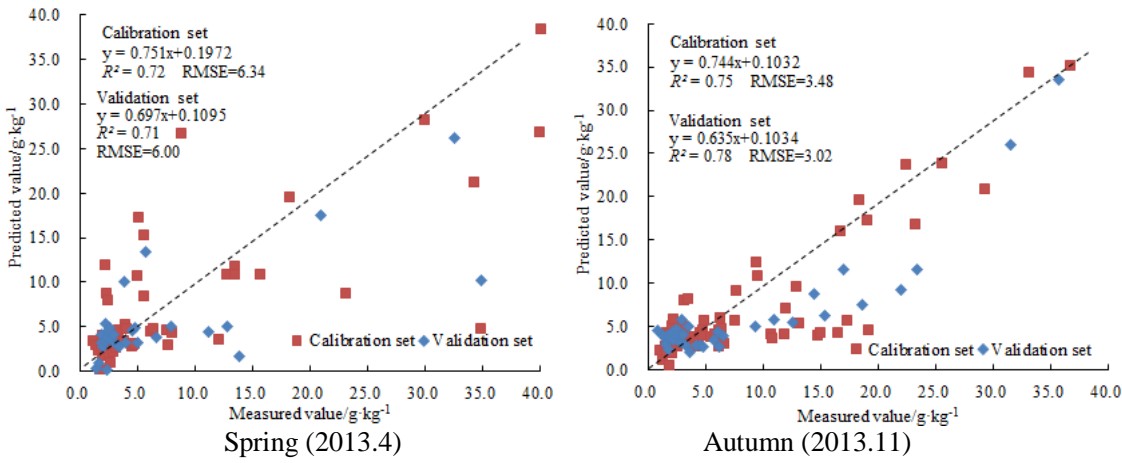

Figure 4. Calibration and validation precision of SSC inversion models in spring and

5    autumn

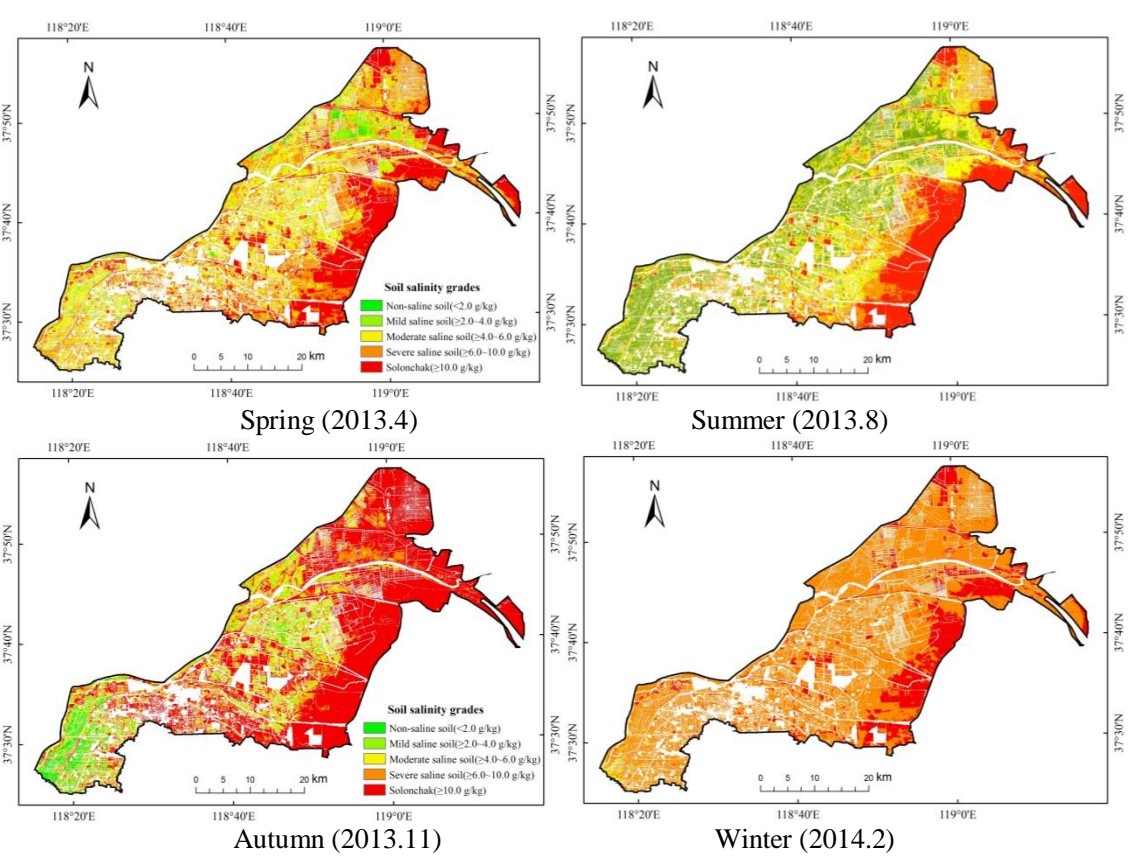

Figure 5. Inversion and distribution of SSC in four seasons