# Peer review of "Monitoring the seasonal dynamics of soil salinization in the Yellow River Delta of China using Landsat data"

_Natural Hazards and Earth System Sciences, 2018_

## Referee Comment (RC1) · Anonymous Referee #1 · 31 Oct 2018

This study compared the field survery and remote sensing image for revealing the distributions of saline soil in Yellow River Delta. The study in this area is very important for crop growth and ecological restoration.

As a whole, this article was well-written and organized. The results were sound and interesting. I think it could be accepted after minor revision.

TitleïijŽ delete "region". Abstract: Why is necessary with distinct seasonal climates? I think some field results could be showed in the abstact. This setence "the SSC optimal model in each season was extracted, then, the spatial distributions and seasonal dynamics of SSC in four seasons were analysed. " was repeated with the second setence. In the introduction, what is the damage of saline soils in Yellow RIver Delta? Figure 1, some labels are not clear. Figure 3, the red underline should be deleted. this distribution pattern is consistent with the results of other studies (Weng et al. 2010; Yang et al. 2015). should be moved to the disscussin part. In figure 5, it seems that the autumn is the most affected saline soil. I think some field results could also be indicated in the conclusion. Most of the last paragraph in disscussion is not really disscussion.
* * *

---

## Referee Comment (RC2) · Anonymous Referee #2 · 11 Nov 2018

The authors used Landsat to monitor seasonal dynamics of soil salinization and compared three regression algorithms for salinity content inversion. Besides, the authors added SWIR to traditional vegetation index. Overall, there was a certain innovation of the manuscript; however, there were some questions and mistakes in this manuscript. Please check it and revise it strictly accordingly.

1. The title is better to revise to 'Monitoring seasonal dynamics of soil salinization in the Yellow River Delta region of China using Landsat data'

2. Please revise the abstract, especially for the part of methods, from your abstract readers could not understand which regression models were used in the prediction model.

3. 'The SSC best inversion model of spring was also determined as the optimal model of winter, similarly, the best model of autumn was also as the optimal model of summer' this sentence was very confusing, please revise it.

4. I am a bit confused about your four seasons inversion. You monitor four seasons dynamics of soil salinization; why didn't you build models for four seasons and just choose two season for models building. You may discuss the soil salinity content of spring was similar with that of winter, for example, while these two seasons vegetation was quite different. So why you applied the spring soil salinity model to that of winter. Please add this discussion.

5. In the part of '2.5 Inversion model construction and optimization', the first six lines were too redundant, please simplify it.

6. Conclusions should be simplified.

7. There were some mistakes in the manuscript, eg, line 6 of abstract, the word should be constructed. Please read carefully and avoid these minor mistakes.

8. The language should be polished before publication.

---

## Author Comment (AC1) · 16 Dec 2018

Thank you for your comments concerning our manuscript which were posted on the NHESS Discussion page on October 31, 2018. Those comments are helpful and constructive for improving our manuscript and future research. The comments and our responses are presented below.

1. This study compared the field survey and remote sensing image for revealing the distributions of saline soil in Yellow River Delta. The study in this area is very important for crop growth and ecological restoration. As a whole, this article was well-written and organized. The results were sound and interesting. I think it could be accepted after

minor revision.

Response: We appreciate the encouraging comments on this study, and we will revise the article carefully.

2. Title:delete "region".

Response: We agree with the comment and the word "region" in the title will be deleted in the revised manuscript.

3. Abstract: Why is necessary with distinct seasonal climates? I think some field results could be showed in the abstract.

Response: In regions with distinct seasons, the difference of rainfall or evaporation is great in different season, the change of soil moisture is obvious, and soil salinity has close relation to the soil moisture, then soil salinity between seasons varies usually greatly. Therefore it is very necessary to monitor seasonal dynamics of soil salinization with distinct seasonal climates. In order to show the variation of soil salt in four seasons, we agree with the comment and will add some field results to the abstract in the revised manuscript.

4. This sentence "the SSC optimal model in each season was extracted, then, the spatial distributions and seasonal dynamics of SSC in four seasons were analysed." was repeated with the second sentence.

Response: We agree with the comment and will delete some repeated information in the revised manuscript.

5. In the introduction, what is the damage of saline soils in Yellow River Delta?

Response: In Yellow River Delta, soil salinization can result in large reduction of agricultural production and fragile ecological environment. We will add some descriptions to the introduction in the revised manuscript about the damage of saline soils in Yellow River Delta.

6. Figure 1, some labels are not clear. Figure 3, the red underline should be deleted.

Response: We agree with the comment, the labels in Figure1will be clear and the red underline in Figure 3 will be deleted in the revised manuscript before resubmission.

7. This distribution pattern is consistent with the results of other studies (Weng et al. 2010; Yang et al. 2015) should be moved to the discussion part.

Response: We agree with the comment, the above-mentioned sentence will be deleted in results and relative discussion will be added to the revised manuscript before resubmission.

8. In figure 5, it seems that the autumn is the most affected saline soil. I think some field results could also be indicated in the conclusion.

Response: From Figure 5 and Table 7, the SSC in autumn was largely dominated by severely saline soil and solonchak (combined proportion of 77.75%); in winter, the SSC was principally severely saline and solonchak, with the combined proportion of 99.19%, of which the severe saline soil contributed 80.71%. Therefore the winter is the most affected saline soil. In order to provide more clarity we agree with the comment and will add some field results to the conclusion in the revised manuscript.

9. Most of the last paragraph in discussion is not really discussion.

Response: The last paragraph of discussion provides the probable reason of the model selection results and the shortage based on data of the time point, some descriptions maybe are too redundant, we agree with the comment and will delete some descriptions of the last discussion paragraph in the revised manuscript.

---

## Author Comment (AC2) · 16 Dec 2018

We are pleased to respond to the helpful and constructive comments, which were posted on the NHESS Discussion page on November 11, 2018. Your comments and our responses are presented below.

1. The title is better to revise to 'Monitoring seasonal dynamics of soil salinization in the Yellow River Delta region of China using Landsat data'

Response: We agree with the comment and the title will be revised to 'Monitoring seasonal dynamics of soil salinization in the Yellow River Delta of China using Landsat

data' in the revised manuscript.

2. Please revise the abstract, especially for the part of methods, from your abstract readers could not understand which regression models were used in the prediction model.

Response: We agree with the comment and will add the description about regression models to the abstract in the revised manuscript.

3. 'The SSC best inversion model of spring was also determined as the optimal model of winter, similarly, the best model of autumn was also as the optimal model of summer' this sentence was very confusing, please revise it.

Response: This experiment indicates that the best inversion model of spring could be applied for the SSC inversion of winter; at the same time, the best inversion model of autumn could also be applied to the SSC inversion of summer in the Yellow River Delta. We agree with the comment and will make changes to the abstract in the revised manuscript.

4. I am a bit confused about your four seasons inversion. You monitor four seasons dynamics of soil salinization; why didn't you build models for four seasons and just choose two season for models building. You may discuss the soil salinity content of spring was similar with that of winter, for example, while these two seasons vegetation was quite different. So why you applied the spring soil salinity model to that of winter. Please add this discussion.

Response: because the SSC is dynamic through the four seasons of a year, applying the same inversion model to analyze the SSC quantitatively in different seasons is not adequate, however, building a model for each season is cumbersome and impractical, so we chose two seasons for models building and studied the models applicability. Based on the data of time point, this experiment indicates the best inversion model of spring can be applied to winter. This model selection results may be due to the short

time intervals, similar soil salt content and climatic conditions between February and April in the Yellow River Delta. In order to respond more accurately to the dynamic changes of soil salt, a period of SSC should be selected as the seasonal salt data, which will be the future research. The last paragraph of discussion provided the relative discussion; in order to provide more clarify we will revise some description in the revised manuscript.

5. In the part of '2.5 Inversion model construction and optimization', the first six lines were too redundant, please simplify it.

Response: We agree with the comment and will simplify the description in the revised manuscript.

6. Conclusions should be simplified.

Response: We agree with the comment and the conclusions will be simplified in the revised manuscript.

7. There were some mistakes in the manuscript, eg, line 6 of abstract, the word should be constructed. Please read carefully and avoid these minor mistakes.

Response: We appreciate your carefulness and we will revise carefully this kind of mistakes in the revised manuscript.

8. The language should be polished before publication.

Response: We agree with the comment and the language will be polished in the revised manuscript.

---

## Referee Comment (RC3) · Anonymous Referee #3 · 31 Jan 2019

General comments The authors aimed to develop models to predict soil salinity over different seasons using an improved vegetation index. However, I found that the drawbacks of the earlier indices were not explained in the Introduction. The Discussion also needs to be improved, particularly the third paragraph since it only repeats the Results.

Specific comments P5 Section 2.2. How did the authors ensure the accuracy of their select months compared to the other months? It is also unclear why the samples collected in spring and autumn were used to develop the inversion models while the samples of winter and summer were used to validate the models. P11L2 What is the relationship between 5 grades of soil salinization and figure 5?

[Figure]

Technical corrections There are numerous convoluted sentences that can be simply
re-written. Some of the most striking examples are: a. Section 2.5. P7L4 So, the
92 samples of spring were divided into . . . . . . . Among the 110 samples of autumn,
74 samples . . . –> Of the 92 samples collected during spring, 62 samples were used
for calibration and the other 30 samples for validation. Similarly, of the 110 samples
collected during autumn,. . . . . . b. Section 3.3.2. Table 5 shows the estimation accuracy.
We can see that the best..... –> Based on the estimation accuracy (Table 5), the best....
c. Section 3.1 Our results showed that SSC reached its maximum concentration in
winter (specify the number) and decreased gradually (Table 2). d. Section 3.2 Table
3 shows that the correlation coefficients between ERVI or EDVI and SSC was very
significant (R2>0.69; P<0.01). Based on these findings, ERVI and EDVI were selected
as the IVI for spring while ENDVI and ERVI were selected as the IVI for autumn.

I strongly suggest the authors to seek professional English proof-reader to help with
the overall language presentation of the manuscript since there are also numerous
grammatical errors. I only list a few of them below.

Abstract L16 constucted –> constructed L23 a gradually increasing trend –> a gradual
increasing trend L24 ... and its seasonal dynamics as that soil salinity accumulates...
–> It also underwent the following seasonal dynamics: .... (Results should be stated in
past tense) Introduction P2L7 ... salinization dynamics is ... –> are P2L20 Extensive re-
searches ... –> studies; research is an uncountable noun and it is not the equivalent of
research papers P2L4 ... obvious geographical advantages –> What are the obvious
geographical advantages of unused land? Please clarify P2L10 ...obvious seasonal
characteristics.... –> distinct seasonal characteristics... P4L1 ...it was applied for map-
ping...and seasonal SSC dynamics were analyzed –> for mapping... and analyzing
seasonal SSC dynamics Materials and Methods P4L8 This area has characteristic –>
This area is characterized by.... P4L10 delete "the levels of soil profile are obvious"
P5L2 replace "the seasonal climate" with "seasonality" P5L14 in situ should be written
in italic P7L2 . . .sorted separately. . . –> separated P7L13 At the same way –>using the

same procedures

---

## Author Comment (AC4) · 27 Feb 2019

Thank you for your comments concerning our manuscript which were posted on the NHESS Discussion page on January 31, 2019. Those comments are helpful and constructive for improving our manuscript and future research. The comments and our responses are presented below.

1. The authors aimed to develop models to predict soil salinity over different seasons using an improved vegetation index. However, I found that the drawbacks of the earlier indices were not explained in the Introduction of the revised manuscript.

[Figure]

Response: We will add some description about the drawbacks of the earlier indices in the Introduction of the revised manuscript.

2. The Discussion also needs to be improved, particularly the third paragraph since it only repeats the Results.

Response: We agree with the comment, the discussion will be revised carefully. The third paragraph mainly discussed the results and disadvantages based on the data of time point. We will delete the repeat description of results in the revised manuscript.

3. How did the authors ensure the accuracy of their select months compared to the other months?

Response: Only based on the data of time point in one season, it is really difficult to ensure the accuracy of the selected time point compared to the other months, so, in order to respond more accurately to the dynamic changes of soil salt, a period of SSC should be selected as the seasonal salt data, which will be the future research. We will add related discussion in the revised manuscript.

4. It is also unclear why the samples collected in spring and autumn were used to develop the inversion models while the samples of winter and summer were used to validate the models.

Response: Firstly, in the YRD regions with distinct seasons, soil salinity between seasons varies usually greatly, it is not appropriate applying the same model to four seasons. Secondly, from the descriptive statistics of the soil samples SSC (Table 2), the SSC in spring is close to winter meanwhile the SSC in summer is close to autumn, so it is feasible to adopt the same model in spring and winter, meanwhile it is feasible to adopt the same model in autumn and summer. Thirdly, in the YRD regions soil salts aggregate to the soil surface in spring, spring is often chosen to study soil salinity inversion (Weng et al., 2010), and because the summer vegetation is too luxuriant, the autumn is more suitable for the study season than summer (Dehni & Lounis, 2012;

Yang et al., 2015), so the samples collected in spring and autumn were used to develop the inversion models while the samples of winter and summer were used to validate the models.

5. P11L2 What is the relationship between 5 grades of soil salinization and figure 5?

Response: 5 grades of soil salinization are non-saline soil, mild saline soil, moderate saline soil, severe saline soil, and solonchak, the degree of soil salinization gradually increased. We will add some description about the relationship between 5 grades of soil salinization in section 3.4.1 of the revised manuscript.

6. There are numerous convoluted sentences that can be simply re-written.

Response: We accept the comment, some of the most striking examples will be revises one by one and some sentences will be re-written, we will revise the whole manuscript carefully.

7. I strongly suggest the authors to seek professional English proof-reader to help with the overall language presentation of the manuscript since there are also numerous grammatical errors.

Response: We accept the comment, we will revise the grammatical errors listed one by one and will seek professional English proof-reader to help.

8. What are the obvious geographical advantages?

Response: The obvious geographical advantages mainly refers to that the Yellow River Delta is located in the junction part of the Beijing-Tianjin-Hebei metropolitan and Shandong Peninsula, and there is a national-level high-efficiency ecological economic region in China. In order to clarify clearly the information, we will modify some description in the revised manuscript.

---

## Author Response (AR1)

**A point-by-point response to the reviews**

**Respond to Referee 1:**

Thank you for your comments concerning our manuscript which were posted on the NHESS Discussion page on October 31, 2018. Those comments are helpful and constructive for improving our manuscript and future research. The comments and our responses are presented below.

1. This study compared the field survey and remote sensing image for revealing the distributions of saline soil in Yellow River Delta. The study in this area is very important for crop growth and ecological restoration.

As a whole, this article was well-written and organized. The results were sound and interesting. I think it could be accepted after minor revision.

**Response:** We appreciate the encouraging comments on this study, and we revised the article carefully.

2. TitleïijŽ delete "region".

**Response:** We agree with the comment and the word "region" in the title was deleted in the revised manuscript. The title was revised as "Monitoring the seasonal dynamics of soil salinization in the Yellow River Delta of China using Landsat data".

3. Abstract: Why is necessary with distinct seasonal climates? I think some field results could be showed in the abstract.

**Response:** In regions with distinct seasons, the difference of rainfall or evaporation is great in different season, the change of soil moisture is obvious, and soil salinity has close relation to the soil moisture, then soil salinity between seasons varies usually greatly. Therefore it is very necessary to monitor seasonal dynamics of soil salinization with distinct seasonal climates. In order to show the variation of soil salt in four seasons, we agree with the comment and added some field results to the abstract in the revised manuscript.

Page 1 Line 19-20, "The results indicated that the SSC varied obviously between seasons in the YRD, and" was added.

4. This sentence "the SSC optimal model in each season was extracted, then, the spatial distributions and seasonal dynamics of SSC in four seasons were analysed." was repeated with the second sentence.

**Response:** We agree with the comment and deleted some repeated information in the revised manuscript.

5. In the introduction, what is the damage of saline soils in Yellow River Delta?

**Response:** In Yellow River Delta, soil salinization can result in large reduction of agricultural production and fragile ecological environment. We added some descriptions to the introduction in the revised manuscript about the damage of saline soils in Yellow River Delta. Page 2 Line 5-6, "Moreover, as a form of land degradation, soil salinization can degrade soil quality and lead to ecosystem risks (Huang et al. 2015; Zhao et al. 2018)." was added.

6. Figure 1, some labels are not clear. Figure 3, the red underline should be deleted.

**Response:** We agree with the comment, the labels in Figure1 was revised to be clear and the red underline in Figure 3 was deleted in the revised manuscript.

7. This distribution pattern is consistent with the results of other studies (Weng et al. 2010; Yang et al. 2015) should be moved to the discussion part.

**Response:** We agree with the comment, the above-mentioned sentence was deleted in results and relative discussion was added to the revised manuscript.

Page 13 Line 11-18, the discussion of "In this work, we introduced the SWIR band and proposed an improved vegetation index to increase the accuracy of SSC inversion models. The spatial distributions of SSC in the four seasons showed similar characteristics. There was a gradually increasing trend of soil salinity from southwest to northeast in the study region, and this distribution pattern is consistent with the results of other studies (Weng et al. 2010; Yang et al. 2015). Weng et al. (2010) also established an SSC remote sensing revision model using the data from 2153 ~ 2254 nm and 1941 ~ 2092 nm in the region of YRD and achieved good results." was added.

8. In figure 5, it seems that the autumn is the most affected saline soil. I think some field results could also be indicated in the conclusion.

**Response:** From Figure 5 and Table 7, the SSC in autumn was largely dominated by severely saline soil and solonchak (combined proportion of 77.75%); in winter, the SSC was principally severely saline and solonchak, with the combined proportion of 99.19%, of which the severe saline soil contributed 80.71%. Therefore the winter is the most affected saline soil. In order to provide more clarity we agree with the comment and added some field results to the conclusion in the revised manuscript.

Page 16 Line 10-11, the statement of "These results are consistent with the results of field sampling, which showed that the SSC is highest in winter, followed by spring and autumn, and lowest in summer." was added.

9. Most of the last paragraph in discussion is not really discussion.

**Response:** The last paragraph of discussion provides the probable reason of the model selection results and the shortage based on data of the time point, some descriptions maybe are too redundant, we agree with the comment and deleted some descriptions of the last discussion paragraph in the revised manuscript.

**Respond to Referee 2:**

We are pleased to respond to the helpful and constructive comments, which were posted on the NHESS Discussion page on November 11, 2018. Your comments and our responses are presented below.

1. The title is better to revise to 'Monitoring seasonal dynamics of soil salinization in the Yellow River Delta region of China using Landsat data'

**Response:** We agree with the comment and the title was revised to 'Monitoring seasonal dynamics of soil salinization in the Yellow River Delta of China using Landsat data' in the revised manuscript.

2. Please revise the abstract, especially for the part of methods, from your abstract readers could not understand which regression models were used in the prediction model.

**Response:** We agree with the comment and added the description about regression models to the abstract in the revised manuscript.

Page 1 Line 12-17, the statement of "The article is to explore the optimal inversion models of soil salinity content (SSC) in different seasons and to achieve the spatial distribution and seasonal dynamics of SSC in Kenli district in the Yellow River Delta (YRD) region of China. Based on the Landsat data in 2013, the improved vegetation indices (IVI) were constructed,

which were then applied in the SSC inversion model construction." was corrected as "The article took Kenli district in the Yellow River Delta (YRD) of China as the experimental area. Based on Landsat data from spring and autumn, improved vegetation indices (IVIs) were created, which were then applied to the inversion modeling of soil salinity content (SSC) by employing stepwise multiple linear regression, back propagation neural network and support vector machine methods.".

3. 'The SSC best inversion model of spring was also determined as the optimal model of winter, similarly, the best model of autumn was also as the optimal model of summer' this sentence was very confusing, please revise it.

**Response:** This experiment indicates that the best inversion model of spring could be applied for the SSC inversion of winter; at the same time, the best inversion model of autumn could also be applied to the SSC inversion of summer in the Yellow River Delta. We agree with the comment and made changes to the abstract in the revised manuscript.

Page 1 Line 21-23, the statement of "The SSC best inversion model of spring was also determined as the optimal model of winter, similarly, the best model of autumn was also as the optimal model of summer." was corrected as "The best SSC inversion model for spring could be applied to the SSC inversion in winter; similarly, the best model for autumn could also be applied to SSC inversion in summer.".

4. I am a bit confused about your four seasons inversion. You monitor four seasons dynamics of soil salinization; why didn't you build models for four seasons and just choose two season for models building. You may discuss the soil salinity content of spring was similar with that of winter, for example, while these two seasons vegetation was quite different. So why you applied the spring soil salinity model to that of winter. Please add this discussion.

**Response:** because the SSC is dynamic through the four seasons of a year, applying the same inversion model to analyze the SSC quantitatively in different seasons is not adequate, however, building a model for each season is cumbersome and impractical, so we chose two seasons for models building and studied the models applicability. Based on the data of time point, this experiment indicates the best inversion model of spring can be applied to winter. This model selection results may be due to the short time intervals, similar soil salt content and climatic conditions between February and April in the Yellow River Delta. In order to respond more accurately to the dynamic changes of soil salt, a period of SSC should be selected as the seasonal salt data, which will be the future research. The last paragraph of discussion provided the relative discussion; in order to provide more clarify we revised some description in the revised manuscript.

Page 15 Line 6-13, the discussion paragraph of "Based on the time point data, the results indicated that the SSC inversion model for spring could be applied to SSC inversion in winter, while the SSC inversion model for autumn could also be applied to SSC inversion in summer in the YRD. These model selection results may be due to the short time intervals, similar soil salt contents and climatic conditions between February and April and between August and November in the YRD. To respond more accurately to the dynamic changes in soil salt, a period of SSC data should be selected as the seasonal salt data, which will be further studied in future research." was added.

5. In the part of '2.5 Inversion model construction and optimization', the first six lines were too redundant, please simplify it.

**Response:** We agree with the comment and simplified the description in the revised manuscript.

Page 7 Line 22-25 and Page8 Line 1-3, the statement of "Two-thirds of the samples were chosen for the calibration set, and the remaining samples were used as the validation set. So, the 92 samples of spring were divided into two groups, specifically one group of 62 samples for calibration and the other group of 30 samples for validation. Among the 110 samples of autumn, 74 samples were used for calibration, and the remaining 36 samples were used for validation." was corrected as "Two-thirds of the samples were chosen for the calibration set, and the remaining samples were used as the validation set. Therefore, of the 92 samples collected during spring, 62 samples were used for calibration, and the other 30 samples were used for validation. Similarly, of the 110 samples collected during autumn, 74 samples were used for calibration, and the other 36 samples were used for validation.".

6. Conclusions should be simplified.

**Response:** We agree with the comment and the conclusions were simplified in the revised manuscript.

Page 15 Line 6-13, the discussion paragraph of "Based on the time point data, the results indicated that the SSC inversion model for spring could be applied to SSC inversion in winter, while the SSC inversion model for autumn could also be applied to SSC inversion in summer in the YRD. These model selection results may be due to the short time intervals, similar soil salt contents and climatic conditions between February and April and between August and November in the YRD. To respond more accurately to the dynamic changes in soil salt, a period of SSC data should be selected as the seasonal salt data, which will be further studied in future research." was added.

Page 15 Line 16-22, the statement of "In this experiment, the results showed that the extended ratio vegetation index (ERVI) and extended difference vegetation index (EDVI) were as IVI of spring, while the extended normalized difference vegetation index (ENDVI) and extended ratio vegetation index (ERVI) were as the IVI of autumn. The best and most stable accuracy of SSC was produced by the SVM models based on the IVI; therefore, these models were selected as the best SCC inversion models of spring and autumn. The experiment results would contribute to the quantitative and accurate monitoring of soil salinization with multispectral imaging." was corrected as "In this experiment, the results showed that the ERVI and EDVI were the IVIs for spring, while the ENDVI and ERVI were the IVIs for autumn. These models based on the IVIs that utilized the SVM method were selected as the best SSC inversion models for spring and autumn. The experimental results contribute to the quantitative and accurate monitoring of soil salinization with multispectral imaging and provide data and technical support for saline soil management and utilization and ecological environment protection.".

7. There were some mistakes in the manuscript, eg, line 6 of abstract, the word should be constructed. Please read carefully and avoid these minor mistakes.

**Response:** We appreciate your carefulness and we revised carefully this kind of mistakes in the revised manuscript.

8. The language should be polished before publication.

**Response:** We agree with the comment and the language was polished in the revised manuscript.

**Respond to Referee 3:**

Thank you for your comments concerning our manuscript which were posted on the NHESS Discussion page on January 31, 2019. Those comments are helpful and constructive for improving our manuscript and future research. The comments and our responses are presented below.

1. The authors aimed to develop models to predict soil salinity over different seasons using an improved vegetation index. However, I found that the drawbacks of the earlier indices were not explained in the Introduction of the revised manuscript.

**Response:** We added some description about the drawbacks of the earlier indices in the Introduction of the revised manuscript.

Page 3 Line 1-13, the whole paragraph of "To a certain extent, information on the damage to vegetation caused by soil salinization can help to determine the degree and trend of soil salinization. Therefore, traditional vegetation indices (VIs), such as the normalized difference vegetation index (NDVI), the ratio vegetation index (RVI), and the difference vegetation index (DVI), can be used as indicators to determine the degree of soil salinization (Elmetwalli et al. 2012; Li et al. 2013; Goto et al. 2015). However, the accuracy of the models based on traditional VIs must be improved (Iqbal 2011). Traditional VIs involve the data from only two bands in the visible and near-infrared region, and there are often significant correlations between traditional VIs (USGS, 2013). Therefore, it is worth studying whether the addition of data from the shortwave infrared band, which has long wavelengths and contains considerable information, can improve the accuracy and stability of soil salinity content (SSC) inversion models." was added

2. The Discussion also needs to be improved, particularly the third paragraph since it only repeats the Results.

**Response:** We agree with the comment, the discussion was revised carefully. The third paragraph mainly discussed the results and disadvantages based on the data of time point. We deleted the repeat description of results in the revised manuscript.

Page 15, the discussion paragraph of "The article aimed to build optimal SSC inversion models for different seasons according to soil salinity condition in one year, and the SSC at the point of time was used as the corresponding seasonal salt data. The results indicated that the SSC inversion model of spring could be applied in winter resulting in $R^2$ of 0.66 and RMSE of 7.57 g/kg, while the SSC inversion model of autumn could also be applied in summer resulting in $R^2$ of 0.65 and RMSE of 3.60 g/kg within a year in the YRD region. This model selection results may be due to the short time intervals and similar soil salt climatic conditions between February and April and between August and November in the Yellow River Delta region. In order to respond more accurately to the dynamic changes of soil salt, a period of SSC should be selected as the seasonal salt data, which will be the future research, and the application of the SSC inversion models for different years will be explored." was deleted.

3. How did the authors ensure the accuracy of their select months compared to the other months?

**Response:** Only based on the data of time point in one season, it is really difficult to ensure the accuracy of the selected time point compared to the other months, so, in order to respond

more accurately to the dynamic changes of soil salt, a period of SSC should be selected as the seasonal salt data, which will be the future research. We added related discussion in the revised manuscript.

Page 15 Line 6-13, the discussion paragraph of "Based on the time point data, the results indicated that the SSC inversion model for spring could be applied to SSC inversion in winter, while the SSC inversion model for autumn could also be applied to SSC inversion in summer in the YRD. These model selection results may be due to the short time intervals, similar soil salt contents and climatic conditions between February and April and between August and November in the YRD. To respond more accurately to the dynamic changes in soil salt, a period of SSC data should be selected as the seasonal salt data, which will be further studied in future research." was added.

4. It is also unclear why the samples collected in spring and autumn were used to develop the inversion models while the samples of winter and summer were used to validate the models.

**Response:** Firstly, in the YRD regions with distinct seasons, soil salinity between seasons varies usually greatly, it is not appropriate applying the same model to four seasons. Secondly, from the descriptive statistics of the soil samples SSC (Table 2), the SSC in spring is close to winter meanwhile the SSC in summer is close to autumn, so it is feasible to adopt the same model in spring and winter, meanwhile it is feasible to adopt the same model in autumn and summer. Thirdly, in the YRD regions soil salts aggregate to the soil surface in spring, spring is often chosen to study soil salinity inversion (Weng et al., 2010), and because the summer vegetation is too luxuriant, the autumn is more suitable for the study season than summer (Dehni & Lounis, 2012; Yang et al., 2015), so the samples collected in spring and autumn were used to develop the inversion models while the samples of winter and summer were used to validate the models.

5. P11L2 What is the relationship between 5 grades of soil salinization and figure 5?

**Response:** 5 grades of soil salinization are non-saline soil, mild saline soil, moderate saline soil, severe saline soil, and solonchak, the degree of soil salinization gradually increased. We added some description about the relationship between 5 grades of soil salinization in section 3.4.1 of the revised manuscript.

Page 11 Line 23-24 and Page 12 Line 1, "nonsaline soil, mild saline soil, moderate saline soil, severe saline soil, and solonchak, with the degree of soil salinization gradually increasing from nonsaline soil to solonchak" was added.

6. There are numerous convoluted sentences that can be simply re-written.

**Response:** We accept the comment, some of the most striking examples were revised one by one and some sentences were re-written, we revised the whole manuscript carefully.

7. I strongly suggest the authors to seek professional English proof-reader to help with the overall language presentation of the manuscript since there are also numerous grammatical errors.

**Response:** We accept the comment, we revised the grammatical errors listed one by one and sought professional English proof-reader to help.

8. What are the obvious geographical advantages?

**Response:** The obvious geographical advantages mainly refers to that the Yellow River Delta is located in the junction part of the Beijing-Tianjin-Hebei metropolitan and Shandong Peninsula, and there is a national-level high-efficiency ecological economic region in China.

In order to clarify clearly the information, we modified some description in the revised manuscript.

Page 3 Line 18-22, the statement of "The Yellow River Delta (YRD) region lies within the efficient ecological economic zone of China. With nearly 550,000 ha of unused land, it has obvious geographical advantages and abundant land resources." was corrected as "The Yellow River Delta (YRD) is located at the junction of the Beijing-Tianjin-Hebei metropolitan area and Shandong Peninsula and lies within the efficient ecological economic zone of China, and this region has obvious geographical advantages. With nearly 550,000 ha of unused land, the land resources in this area are rich.".

**Respond to editor:**

We would like to thank you for the in-depth comments and suggestions of our paper. You helped us to stay in line with the special issue and improve key points of the background and make the discussion and conclusion more valuable. The comments and our responses are presented below.

1. I suggest the authors should be think over the hazards/risk topic, and must stay in line with the special issue goals please.

**Response:** As a form of land degradation, soil salinization can degrade soil quality and lead to ecosystem risks, and as the main risk to farmland ecosystems in the Yellow River Delta, soil salinization can result in a large reduction in agricultural production and fragile ecological environments. In the revised manuscript, we added some description to the first and fourth paragraph of introduction and the first paragraph of conclusion.

Page 2 Line 5-6, "Moreover, as a form of land degradation, soil salinization can degrade soil quality and lead to ecosystem risks (Huang et al. 2015; Zhao et al. 2018)." was added. Page 4 Line 1-4, the statement of "which has seriously affected the utilization of land resources as well as the development of the regional economy and society (Yang et al. 2015; Weng et al. 2010)." was corrected as "As the main risk to farmland ecosystems in this region, soil salinization can result in a large reduction in agricultural production and fragile ecological environments; and, soil salinization seriously affects the utilization of land resources as well as the development of the regional economy and society (Yang et al. 2015; Weng et al. 2010).". Page 15 Line 16-22, the statement of "In this experiment, the results showed that the extended ratio vegetation index (ERVI) and extended difference vegetation index (EDVI) were as IVI of spring, while the extended normalized difference vegetation index (ENDVI) and extended ratio vegetation index (ERVI) were as the IVI of autumn. The best and most stable accuracy of SSC was produced by the SVM models based on the IVI; therefore, these models were selected as the best SCC inversion models of spring and autumn. The experiment results would contribute to the quantitative and accurate monitoring of soil salinization with multispectral imaging." was corrected as "In this experiment, the results showed that the ERVI and EDVI were the IVIs for spring, while the ENDVI and ERVI were the IVIs for autumn. These models based on the IVIs that utilized the SVM method were selected as the best SSC inversion models for spring and autumn. The experimental results contribute to the quantitative and accurate monitoring of soil salinization with multispectral imaging and provide data and technical support for saline soil management and utilization and ecological environment protection.".

2. As for the manuscript, the Introduction is short to show the frontiers on remote sensed SSC. Why the seasonal inversion is urgently needed has not been well explained.

**Response:** We agree with the comment, some description about the frontiers on and the drawbacks of the earlier spectra indices used to monitoring soil salt was added in the third paragraph of the introduction and the reason why the seasonal inversion of soil salt content is urgently needed was added in the fourth paragraph of the introduction in the revised manuscript.

Page 3 Line 1-13, the whole paragraph of "To a certain extent, information on the damage to vegetation caused by soil salinization can help to determine the degree and trend of soil salinization. Therefore, traditional vegetation indices (VIs), such as the normalized difference vegetation index (NDVI), the ratio vegetation index (RVI), and the difference vegetation index (DVI), can be used as indicators to determine the degree of soil salinization (Elmetwalli et al. 2012; Li et al. 2013; Goto et al. 2015). However, the accuracy of the models based on traditional VIs must be improved (Iqbal 2011). Traditional VIs involve the data from only two bands in the visible and near-infrared region, and there are often significant correlations between traditional VIs (USGS, 2013). Therefore, it is worth studying whether the addition of data from the shortwave infrared band, which has long wavelengths and contains considerable information, can improve the accuracy and stability of soil salinity content (SSC) inversion models." was added. Page 4 Line 5-9, the statement of "Moreover, the SSC in the YRD region has obvious seasonal characteristics because of the seasonal climate. Real-time, continuous monitoring of soil salinization is particularly important in this region." was corrected as "Moreover, in regions with distinct seasons, the differences in rainfall and evaporation between different seasons are great, the changes in soil moisture are obvious, and soil salinity exhibits a close relation with soil moisture; thus, the soil salinity usually varies greatly between seasons. Therefore, it is particularly necessary to monitor the seasonal dynamics of soil salinization in this region.".

3. In the Discussion, literature comparison with former studies should be added.

**Response:** we agree with the comment and added some comparison with former studies to the first and third paragraph of the discussion in the revised manuscript.

Page 13 Line 13-18, the discussion of "There was a gradually increasing trend of soil salinity from southwest to northeast in the study region, and this distribution pattern is consistent with the results of other studies (Weng et al. 2010; Yang et al. 2015). Weng et al. (2010) also established an SSC remote sensing revision model using the data from 2153 ~ 2254 nm and 1941 ~ 2092 nm in the region of YRD and achieved good results." was added. Page 15 Line 2-4, the statement of "Lu et al. (2016) presented that SSC exhibits seasonal variation in the YRD and that the SSC in spring was higher than that in autumn in the Kenli district, which is consistent with our results." was added.

4. It seems that there are too many tables, it may be better to remain no more than 4 tables in the manuscript.

**Response:** We agree with the comment, Table 1 was deleted and some relevant description was added to the section 2.4; Table 3 was deleted and some relevant description was added to the section 3.2; Table 2 and Table 6 were merged, so there are remain 4 tables in the revised manuscript.

5. Besides, the language should be polished by a native English speaker.

**Response:** We agree with the comment and sought a native English speaker to help.

**Respond to Short Comment 1:**
Thank you for your comments concerning our manuscript which were posted on the NHESS Discussion page on October 31, 2018. Those comments are helpful and constructive for improving our manuscript and future research. The comments and our responses are presented below.

1. This study compared the field survey and remote sensing image for revealing the distributions of saline soil in Yellow River Delta. The study in this area is very important for crop growth and ecological restoration.

As a whole, this article was well-written and organized. The results were sound and interesting. I think it could be accepted after minor revision.

**Response:** We appreciate the encouraging comments on this study, and we revised the article carefully.

2. TitleïïjŽ delete "region".

**Response:** We agree with the comment and the word "region" in the title was deleted in the revised manuscript. The title was revised as "Monitoring the seasonal dynamics of soil salinization in the Yellow River Delta of China using Landsat data".

3. Abstract: Why is necessary with distinct seasonal climates? I think some field results could be showed in the abstract.

**Response:** In regions with distinct seasons, the difference of rainfall or evaporation is great in different season, the change of soil moisture is obvious, and soil salinity has close relation to the soil moisture, then soil salinity between seasons varies usually greatly. Therefore it is very necessary to monitor seasonal dynamics of soil salinization with distinct seasonal climates. In order to show the variation of soil salt in four seasons, we agree with the comment and added some field results to the abstract in the revised manuscript.

Page 1 Line 19-20, "The results indicated that the SSC varied obviously between seasons in the YRD, and" was added.

4. This sentence "the SSC optimal model in each season was extracted, then, the spatial distributions and seasonal dynamics of SSC in four seasons were analysed." was repeated with the second sentence.

**Response:** We agree with the comment and deleted some repeated information in the revised manuscript.

5. In the introduction, what is the damage of saline soils in Yellow River Delta?

**Response:** In Yellow River Delta, soil salinization can result in large reduction of agricultural production and fragile ecological environment. We added some descriptions to the introduction in the revised manuscript about the damage of saline soils in Yellow River Delta.

Page 2 Line 5-6, "Moreover, as a form of land degradation, soil salinization can degrade soil quality and lead to ecosystem risks (Huang et al. 2015; Zhao et al. 2018)." was added.

6. Figure 1, some labels are not clear. Figure 3, the red underline should be deleted.

**Response:** We agree with the comment, the labels in Figure1 was revised to be clear and the red underline in Figure 3 was deleted in the revised manuscript.

7. This distribution pattern is consistent with the results of other studies (Weng et al. 2010; Yang et al. 2015) should be moved to the discussion part.

**Response:** We agree with the comment, the above-mentioned sentence was deleted in results and relative discussion was added to the revised manuscript.

Page 13 Line 11-18, the discussion of "In this work, we introduced the SWIR band and proposed an improved vegetation index to increase the accuracy of SSC inversion models. The spatial distributions of SSC in the four seasons showed similar characteristics. There was a gradually increasing trend of soil salinity from southwest to northeast in the study region, and this distribution pattern is consistent with the results of other studies (Weng et al. 2010; Yang et al. 2015). Weng et al. (2010) also established an SSC remote sensing revision model using the data from 2153 ~ 2254 nm and 1941 ~ 2092 nm in the region of YRD and achieved good results." was added.

8. In figure 5, it seems that the autumn is the most affected saline soil. I think some field results could also be indicated in the conclusion.

**Response:** From Figure 5 and Table 7, the SSC in autumn was largely dominated by severely saline soil and solonchak (combined proportion of 77.75%); in winter, the SSC was principally severely saline and solonchak, with the combined proportion of 99.19%, of which the severe saline soil contributed 80.71%. Therefore the winter is the most affected saline soil. In order to provide more clarity we agree with the comment and added some field results to the conclusion in the revised manuscript.

Page 16 Line 10-11, the statement of "These results are consistent with the results of field sampling, which showed that the SSC is highest in winter, followed by spring and autumn, and lowest in summer." was added.

9. Most of the last paragraph in discussion is not really discussion.

**Response:** The last paragraph of discussion provides the probable reason of the model selection results and the shortage based on data of the time point, some descriptions maybe are too redundant, we agree with the comment and deleted some descriptions of the last discussion paragraph in the revised manuscript.

**A list of all relevant changes made in the manuscript**

1. Page 1 Line 1-2, the title was revised as "Monitoring the seasonal dynamics of soil salinization in the Yellow River Delta of China using Landsat data".

2. Page 1 Line 12-17, the statement of "The article is to explore the optimal inversion models of soil salinity content (SSC) in different seasons and to achieve the spatial distribution and seasonal dynamics of SSC in Kenli district in the Yellow River Delta (YRD) region of China. Based on the Landsat data in 2013, the improved vegetation indices (IVI) were constructed, which were then applied in the SSC inversion model construction." was corrected as "The article took Kenli district in the Yellow River Delta (YRD) of China as the experimental area. Based on Landsat data from spring and autumn, improved vegetation indices (IVIs) were created, which were then applied to the inversion modeling of soil salinity content (SSC) by employing stepwise multiple linear regression, back propagation neural network and support vector machine methods.".

3. Page 1 Line 19-20, "The results indicated that the SSC varied obviously between seasons in the YRD, and" was added.

4. Page 1 Line 21-23, the statement of "The SSC best inversion model of spring was also determined as the optimal model of winter, similarly, the best model of autumn was also as the optimal model of summer." was corrected as "The best SSC inversion model for spring could be applied to the SSC inversion in winter; similarly, the best model for autumn could also be applied to SSC inversion in summer.".

5. Page 1 Line 25, the statement of "its seasonal dynamics were as that" was corrected as "it also underwent the following seasonal dynamics:".

6. Page 2 Line 5-6, "Moreover, as a form of land degradation, soil salinization can degrade soil quality and lead to ecosystem risks (Huang et al. 2015; Zhao et al. 2018)." was added.

7. Page 2 Line 10, "then help to reduce ecological risks" was added.

8. Page 2 Line 16-19, The statement of "Due to the low cost and the ability to map extreme surface expressions of salinity of the imagery, multispectral satellite data, such as Landsat, System Probatoired' Observation dela Terre (SPOT), IKONOS, QuickBird, and the Indian Remote Sensing (IRS) series of satellites, have been used for mapping and monitoring of soil salinity and other properties" was corrected as "Multispectral satellite data, such as Landsat, SPOT, IKONOS, QuickBird, and the Indian Remote Sensing (IRS) series of satellites, have been used to map and monitor soil salinity and other properties due to the low cost and the ability to map extreme surface expressions of salinity".

9. Page 3 Line 1-13, the whole paragraph of "To a certain extent, information on the damage to vegetation caused by soil salinization can help to determine the degree and trend of soil salinization. Therefore, traditional vegetation indices (VIs), such as the normalized difference vegetation index (NDVI), the ratio vegetation index (RVI), and the difference vegetation index (DVI), can be used as indicators to determine the degree of soil salinization (Elmetwalli et al. 2012; Li et al. 2013; Goto et al. 2015). However, the accuracy of the models based on traditional VIs must be improved (Iqbal 2011). Traditional VIs involve the data from only two bands in the visible and near-infrared region, and there are often significant correlations between traditional VIs (USGS, 2013). Therefore, it is worth studying whether the addition of data from the shortwave infrared band, which has long wavelengths and contains considerable

information, can improve the accuracy and stability of soil salinity content (SSC) inversion models." was added to descript the frontiers on and the drawbacks of the earlier spectra indices used to monitoring soil salt.

10. Page 3 Line 18-22, the statement of "The Yellow River Delta (YRD) region lies within the efficient ecological economic zone of China. With nearly 550,000 ha of unused land, it has obvious geographical advantages and abundant land resources." was corrected as "The Yellow River Delta (YRD) is located at the junction of the Beijing-Tianjin-Hebei metropolitan area and Shandong Peninsula and lies within the efficient ecological economic zone of China, and this region has obvious geographical advantages. With nearly 550,000 ha of unused land, the land resources in this area are rich.".

11. Page 4 Line 1-4, the statement of "which has seriously affected the utilization of land resources as well as the development of the regional economy and society (Yang et al. 2015; Weng et al. 2010)." was corrected as "As the main risk to farmland ecosystems in this region, soil salinization can result in a large reduction in agricultural production and fragile ecological environments; and, soil salinization seriously affects the utilization of land resources as well as the development of the regional economy and society (Yang et al. 2015; Weng et al. 2010).".

12. Page 4 Line 5-9, the statement of "Moreover, the SSC in the YRD region has obvious seasonal characteristics because of the seasonal climate. Real-time, continuous monitoring of soil salinization is particularly important in this region." was corrected as "Moreover, in regions with distinct seasons, the differences in rainfall and evaporation between different seasons are great, the changes in soil moisture are obvious, and soil salinity exhibits a close relation with soil moisture; thus, the soil salinity usually varies greatly between seasons. Therefore, it is particularly necessary to monitor the seasonal dynamics of soil salinization in this region.".

13. Page 4 Line 15-16, the statement of "Specifically, we bulit the vegetation indices by introducing the additional data of the short-wave infrared band (SWIR) available in Landsat data." was corrected as "Specifically, VIs were constructed by introducing data from the shortwave infrared band (SWIR) of Landsat data.".

14. Page 5 Line 8-10, the statement of "the levels of soil profile are obvious with salt accumulating in the soil surface while relatively little and well-distributed in the middle and lower part of the soil profile (below the core soil)." was corrected as "salt accumulates on the soil surface, and salt is relatively rare and well-distributed in the middle and lower parts of the soil profile (below the core soil).".

15. Page 6 Line 3-5, the statement of "during August 14~15, 2013, 30 summer samples were collected; 110 autumn samples were collected during November 9~ 13, 2013, and 56winter samples were collected during February 26~ 29, 2014." was corrected as "30 summer samples were collected from August 14-15, 2013; 110 autumn samples were collected from November 9-13, 2013; and 56 winter samples were collected from February 26-29, 2014.".

16. Page 7 Line 1, the statement of "from the Exelis Visual Information Solutions" about the ENVI 5.1 software manufacturer was added.

17. Page 7 Line 13-16, the statement of "including normalized difference vegetation index (NDVI), difference vegetation index (DVI), and ratio vegetation index (RVI) which was shown in Table 1." was corrected as "including the extended normalized difference

vegetation index (ENDVI, (NIR+SWIR-R)/(NIR+SWIR+R)), extended difference vegetation index (EDVI, NIR+SWIR-R), and extended ratio vegetation index (ERVI, (NIR+SWIR)/R). The SWIR band refers to either of the two SWIR bands." and Table 1 was deleted.

18. Page 7 Line 22-25 and Page8 Line 1-3, the statement of "Two-thirds of the samples were chosen for the calibration set, and the remaining samples were used as the validation set. So, the 92 samples of spring were divided into two groups, specifically one group of 62 samples for calibration and the other group of 30 samples for validation. Among the 110 samples of autumn, 74 samples were used for calibration, and the remaining 36 samples were used for validation." was corrected as "Two-thirds of the samples were chosen for the calibration set, and the remaining samples were used as the validation set. Therefore, of the 92 samples collected during spring, 62 samples were used for calibration, and the other 30 samples were used for validation. Similarly, of the 110 samples collected during autumn, 74 samples were used for calibration, and the other 36 samples were used for validation.".

19. Page 8 Line 6-10 and Page8 Line 1-3, the statement of "At the same way, the best model of SSC for autumn was built on the IVI and selected. Finally the best model for spring and autumn was respectively decided and applied to the summer and winter data, then the optimal inversion models of SSC according to soil salinization conditions in different seasons were selected." was corrected as "Using the same procedures, the SSC models for autumn were built on the IVIs, and the best model was selected. Finally, the best models for spring and autumn were selected and applied to the summer and winter data, and then the optimal SSC inversion models according to the soil salinization conditions in different seasons were selected.".

20. Page 9 Line 21-22 and Page 10 Line 1-2, the statement of "The correlation coefficients between the EVI and the SSC of the soil samples are shown in Table 3." was corrected as "In spring, the correlation coefficients between the EVIs and the SSC of the soil samples were -0.52 for ENDVI, -0.69 for ERVI and -0.70 for EDVI; similarly in autumn, the correlation coefficients between the EVIs and the SSC of the soil samples were -0.73 for ENDVI, -0.69 for ERVI and -0.69 for EDVI." and Table 3 was deleted.

21. Page 10 Line 4-5, the statement of "From Table 3, we can see that the correlation between the ERVI or EDVI and the SSC was very significant with the correlation coefficient above 0.69. So the ERVI and EDVI were selected as the IVI for spring, at the same way the ENDVI and ERVI were selected as the IVI for autumn." was corrected as "The results show that the correlation coefficients between the ERVI or EDVI and SSC were very significant ($R^2$>0.69; P<0.01) in spring. Based on these findings, ERVI and EDVI were selected as the IVIs for spring, while ENDVI and ERVI were selected as the IVIs for autumn.".

22. Page 10 Line 10, the headline of "3.3 The best inversion models of SSC and its application in different seasons" was corrected as "3.3 The best SSC inversion models and their application to different seasons".

23. Page 10 Line 11, the headline of "3.3.1 Inversion models of SSC with VI and IVI" was corrected as "3.3.1 SSC inversion models with VIs and IVIs".

24. Page 10 Line 12-15, the statement of "Results of the SSC inversion models in spring based on the IVI are shown in Table 4. In comparing the performance of three modelling methods, the prediction accuracy of the SVM models was the highest followed by the BPNN models, and the SMLR models had the lowest accuracy." was corrected as "The results of the

SSC inversion models in spring based on the IVIs are shown in Table 2. The performances of the three modeling methods were compared, which showed that the prediction accuracy of the SVM models was the highest followed by the BPNN models, and the SMLR models had the lowest accuracy." and Table 2 and Table 6 were merged.

25. Page 11 Line 3-4, the statement of "Table 5 shows the estimation accuracy" was corrected as "Based on the estimation accuracy (Table 3)".

26. Page 11 Line 16-18, the statement of "The descriptive statistics of the inversed SSC in four seasons are shown in Table 6, which are close to those of the collected samples (Table 2)" was corrected as "The descriptive statistics of the inversed SSC in four seasons are shown in the lower half of Table 1, which are close to those of the collected samples (the upper half of Table 1)".

27. Page 11 Line 23-24 and Page 12 Line 1, "nonsaline soil, mild saline soil, moderate saline soil, severe saline soil, and solonchak, with the degree of soil salinization gradually increasing from nonsaline soil to solonchak" was added.

28. Page 12 Line 2-4, the statement of "The distribution of soil salinity grades in the four seasons were mapped (Fig. 5). The spatial distributions of SSC in the four seasons showed similar characteristics. There was a gradually increasing trend of soil salinity from the south-west to the north-east part of the study region, and this distribution pattern is consistent with the results of other studies (Weng et al. 2010; Yang et al. 2015)." was corrected as "The distributions of the soil salinity grades in the four seasons were mapped (Fig. 5) and showed similar characteristics. There was a gradually increasing trend of soil salinity from southwest to northeast in the study region.".

29. Page 13 Line 11-18, the discussion of "In this work, we introduced the SWIR band and proposed an improved vegetation index to increase the accuracy of SSC inversion models. The spatial distributions of SSC in the four seasons showed similar characteristics. There was a gradually increasing trend of soil salinity from southwest to northeast in the study region, and this distribution pattern is consistent with the results of other studies (Weng et al. 2010; Yang et al. 2015). Weng et al. (2010) also established an SSC remote sensing revision model using the data from 2153 ~ 2254 nm and 1941 ~ 2092 nm in the region of YRD and achieved good results." was added.

30. Page 15 Line 2-4, the statement of "Lu et al. (2016) presented that SSC exhibits seasonal variation in the YRD and that the SSC in spring was higher than that in autumn in the Kenli district, which is consistent with our results." was added.

31. Page 15, the discussion paragraph of "The article aimed to build optimal SSC inversion models for different seasons according to soil salinity condition in one year, and the SSC at the point of time was used as the corresponding seasonal salt data. The results indicated that the SSC inversion model of spring could be applied in winter resulting in $R^2$ of 0.66 and RMSE of 7.57 g/kg, while the SSC inversion model of autumn could also be applied in summer resulting in $R^2$ of 0.65 and RMSE of 3.60 g/kg within a year in the YRD region. This model selection results may be due to the short time intervals and similar soil salt climatic conditions between February and April and between August and November in the Yellow River Delta region. In order to respond more accurately to the dynamic changes of soil salt, a period of SSC should be selected as the seasonal salt data, which will be the future research, and the application of the SSC inversion models for different years will be explored." was

deleted.

32. Page 15 Line 2-4, the statement of "Lu et al. (2016) presented that SSC exhibits seasonal variation in the YRD and that the SSC in spring was higher than that in autumn in the Kenli district, which is consistent with our results." was added.

33. Page 15 Line 6-13, the discussion paragraph of "Based on the time point data, the results indicated that the SSC inversion model for spring could be applied to SSC inversion in winter, while the SSC inversion model for autumn could also be applied to SSC inversion in summer in the YRD. These model selection results may be due to the short time intervals, similar soil salt contents and climatic conditions between February and April and between August and November in the YRD. To respond more accurately to the dynamic changes in soil salt, a period of SSC data should be selected as the seasonal salt data, which will be further studied in future research." was added.

34. Page 15 Line 16-22, the statement of "In this experiment, the results showed that the extended ratio vegetation index (ERVI) and extended difference vegetation index (EDVI) were as IVI of spring, while the extended normalized difference vegetation index (ENDVI) and extended ratio vegetation index (ERVI) were as the IVI of autumn. The best and most stable accuracy of SSC was produced by the SVM models based on the IVI; therefore, these models were selected as the best SCC inversion models of spring and autumn. The experiment results would contribute to the quantitative and accurate monitoring of soil salinization with multispectral imaging." was corrected as "In this experiment, the results showed that the ERVI and EDVI were the IVIs for spring, while the ENDVI and ERVI were the IVIs for autumn. These models based on the IVIs that utilized the SVM method were selected as the best SSC inversion models for spring and autumn. The experimental results contribute to the quantitative and accurate monitoring of soil salinization with multispectral imaging and provide data and technical support for saline soil management and utilization and ecological environment protection.".

35. Page 16 Line 7-11, the statement of "The SSC spatial distribution of each season in the study area was determined using the SCC inversion model optimized for each season. In the Yellow River Delta region, the spatial distribution of SSC shows a gradually increasing trend from the south-west to the north-east. The seasonal dynamics of SSC are such that soil salts accumulate in spring, decrease in summer, rise in autumn, and peak in winter." was corrected as "In the YRD region, the spatial distribution of SSC shows a gradually increasing trend from southwest to northeast. The seasonal dynamics of SSC are such that soil salts accumulate in spring, decrease in summer, increase in autumn, and peak in winter. These results are consistent with the results of field sampling, which showed that the SSC is highest in winter, followed by spring and autumn, and lowest in summer.".

36. Page 17, Line 26-32, the references of "Elmetwalli, A. M. H., A. N. Tyler, P. D. Hunter, and C. A.: Salt Detecting and Distinguishing Moisture-and Salinity-induced Stress in Wheat and Maize through in Situ Spectroradiometry Measurements. Remote Sensing Letter 3: 363–372. DOI: 10.1080/01431161.2011.599346, 2012." and "Goto, K. , Goto, T. , Nmor, J. C. , Minematsu, K. , and Gotoh, K. . Evaluating Salinity Damage to Crops Through Satellite Data Analysis: Application to Typhoon Affected Areas of Southern Japan. Natural Hazards, 75(3), 2815-2828. doi: 10.1007/s11069-014-1465-0, 2015." were added.

37. Page 18, Line 16-23, the references of "Li, P., L. Jiang, and Z. Feng.: Cross-comparison

of Vegetation Indices Derived from Landsat-7 Enhanced Thematic Mapper Plus (ETM) and Landsat-8 Operational Land Imager (OLI) Sensors. Remote Sensing 6: 310–329. doi: 10.3390/rs6010310, 2013." and "Lu, Q. , Bai, J. , Fang, H. , Wang, J. , Zhao, Q. , and Jia, J.: Spatial and Seasonal Distributions of Soil Sulfur in Two Marsh Wetlands with Different Flooding Frequencies of the Yellow River Delta, China. Ecological Engineering, 96(96), 63-71. doi: 10.1016/j.ecoleng.2015.10.033, 2016." were added.

38. Page 20, Line 13-16, the reference of "Yang, S.Q., Zhao, W.W., Liu, Y.X., Wang, S., Wang, J., Zhai, R.J.: Influence of Land Use Change on the Ecosystem Service Trade-offs in the Ecological Restoration Area: Dynamics and Scenarios in the Yanhe Watershed, China. Science of the Total Environment, 644: 556–566. doi: 10.1016/j.scitotenv.2018.06.348, 2018." was added.

39. Page 20, Line 21-23, the reference of "Zhao, W.W., Wei, H., Jia, L.Z., Zhang, X., Liu, Y.X.: Soil Erodibility and its Influencing Factors on the Loess Plateau of China: A Case Study in the Ansai Watershed. Solid Earth, 9, 1507–1516, 2018." was added.

40. Page 21, Line 2, the title of Table 1 was corrected as "Table 1. SSC descriptive statistics of samples and inversion".

41. Page 22, Figure 1 was revised to be clear.

42. Page 22, Figure 3, the red underline was deleted.

43. Page 24, the cross heading of Figure 4 was formatted.

44. The whole manuscript was revised by a professional English proof-reader.

The marked-up manuscript version

[revised manuscript text omitted]

---

## Author Response (AR2)

**A point-by-point response to the reviews**

I found a number long sentences which make them very difficult to follow, in addition to the associated grammatical errors. These sentences blurred the message that the authors tried to convey. On an occasion, it is also contrasting to each other, for example: salt accumulates on the soil surface, and salt is relatively rare and well-distributed in the middle and lower parts of the soil profile (below the core soil). More examples for these type of sentences can be found in the following sections:

**Response:** Thanks for the helpful comments from Report #2, we revised carefully the manuscript, especially the long sentences. A point-by-point response is as follows.

1. P2: L6-10 Obtaining information on soil characteristics, such as the degree of salinity and geographical distribution of saline soil in real time, and improving our ability to forecast soil salinization dynamics are necessary prerequisites for scientific management, reasonable improvement and utilization of regional saline soil, then help to reduce ecological risks (Melendez-Pastor et al. 2012; Yang et al. 2018).

**Response:** This sentence was corrected to "Therefore, the scientific treatment and utilization of saline soil are of great significance to regional agricultural production and ecological security. Moreover, it is necessary prerequisite to obtain the degree, geographical distribution and dynamics of soil salinization in real time (Melendez-Pastor et al. 2012; Yang et al. 2018).".

2、P4: L1-5 As the main risk to farmland ecosystems in this region, soil salinization can result in a large reduction in agricultural production and fragile ecological environments; and, soil salinization seriously affects the utilization of land resources as well as the development of the regional economy and society (Yang et al. 2015; Weng et al. 2010).

**Response:** This sentence was corrected to "As the main risk to farmland ecosystems in this region, soil salinization can result in large reductions in agricultural and fragile ecological environments, which could influence the development of the regional economy and society (Yang et al. 2015; Weng et al. 2010).".

3、P4: L5-8 Moreover, in regions with distinct seasons, the differences in rainfall and evaporation between different seasons are great, the changes in soil moisture are obvious, and soil salinity exhibits a close relation with soil moisture; thus, the soil salinity usually varies greatly between seasons.

**Response:** This sentence was corrected to "Nevertheless, in regions with distinct seasons, the changes in soil moisture are obvious due to the great differences in rainfall and evaporation between different seasons. Thus, because soil salinity is closely related to soil moisture, soil salinity usually varies greatly between seasons.".

4、P5: L7-10 The soil parent material is the Yellow River alluvial material, and the soil texture is light; salt accumulates on the soil surface, and salt is relatively rare and well-distributed in the middle and lower parts of the soil profile (below the core soil).

**Response:** This sentence was corrected to "The soil parent material is Yellow River alluvial material, and the soil texture is light. The salt in groundwater can easily reach the soil surface with the evaporation of water from the soil; thus, salt accumulates on

the soil surface while it is relatively rare in the middle and lower parts of the soil profile (below the core soil). ".

5、P9: L14-L16 As determined from the minimum, maximum, and mean values, our results showed that the SSC reached its maximum concentration in winter (the mean = 9.50 g/kg) and decreased gradually, and the SSC varied obviously between seasons.

**Response:** This sentence was corrected to "As determined from the minimum, maximum, and mean values, the SSC reached its maximum concentration in winter (the mean = 9.50 g/kg) and varied obviously between seasons. As the coefficients of variation for all four seasons were greater than 1.00, the overall SSC gradient was obvious, especially in winter and spring.".

6、P10: L13-15 The performances of the three modeling methods were compared, which showed that the prediction accuracy of the SVM models was the highest followed by the BPNN models, and the SMLR models had the lowest accuracy.

**Response:** This sentence was corrected to "The performances of the three modeling methods were compared, which indicated that the SVM models had the highest prediction accuracy, followed by the BPNN models, and the SMLR models had the lowest accuracy.".

7、P11: L22-P12: L1 According to the classification standard of coastal saline soil in the semihumid area of China, the study area was divided into 5 grades of soil salinization: nonsaline soil, mild saline soil, moderate saline soil, severe saline soil, and solonchak, with the degree of soil salinization gradually increasing from nonsaline soil to solonchak.

**Response:** This sentence was corrected to "According to the classification standard of coastal saline soil in the semihumid area of China, the study area was divided into 5 grades: nonsaline soil, mild saline soil, moderate saline soil, severe saline soil, and solonchak.".

Other comments:

1、The Abstract is started with an opinion (It is necessary to monitor the seasonal dynamics of soil salinization in regions with distinct seasonal climates). However, it has no introduction whatsoever which make me question: Why is it necessary?

**Response:** We agree with the comment and added some information to the abstract; P1 Lines 11-13.

2、Please use past tense to describe the result, including in all sections of the manuscript. I still found numerous grammatical errors and awkward sentences throughout the manuscript.

**Response:** We agree with the comment and used past tense to describe the results in the revised manuscript.

3、P2: Please move L11-L14 to the beginning of the next paragraph

**Response:** We agree with the comment and moved the relative descriptions to the beginning of the next paragraph, P2 Lines 13-16.

4、P3: L8 …. there are often significant correlations between traditional VIs (USGS, 2013). I assume it is a drawback for using the traditional VIs. However, I believe a little bit of explanation on why significant correlations may be a disturbance will be helpful for the readers.

**Response:** We agree with the comment and added some explanations on P3 Lines 8-9.

5、P3: Second paragraph is a very convoluted paragraph. I suggest to separate explanation regarding location and season rather than mix them.

**Response:** We agree with the comment and separated the paragraph regarding the seasons and location into two paragraphs: the last paragraph on P3 and the first paragraph on P4.

6、P9: L18-19 The standard deviation and coefficient of variation showed that the SSC gradient was significant overall, especially in winter and spring. How did you differentiate between significant and non-significant results?

**Response:** The standard deviation and coefficient of variation reflect the degree of data dispersion; the greater the coefficient of variation is, the higher the degree of dispersion. To avoid misunderstanding, this sentence was revised to "As the coefficients of variation for all four seasons were greater than 1.00, the overall SSC gradient was obvious, especially in winter and spring." on P9 Lines 16-18.

7、P13: L16-18 Weng et al. (2010) also established an SSC remote sensing revision model using the data from 2153 ~ 2254 nm and 1941 ~ 2092 nm in the region of YRD and achieved good results. What do you mean 'good results'? Please clarify.

**Response:** We agree with the comment and added some data to clarify the "good results" on P13 Lines 13-15.

8、P14: L3-6 In autumn, rainfall and temperature are reduced, and there is little coverage of natural vegetation. At this time, cotton has been collected, and only withered cotton leaves and rods remain in the field and wheat has just begun to emerge out of the soil. I believe that cotton and wheat are categorized as crops instead of natural vegetation in the Method section.

**Response:** We agree with the comment and some of the relative descriptions were revised on P14 Lines 1-3.

Editor Decision: Publish subject to minor revisions (review by editor) (28 Apr 2019) by Wenwu Zhao
Comments to the Author:
This topic is interesting, and I am glad to notice that the manuscript has been improved. While, just as the comments from Report #2, more efforts should be paid on the manuscript and the language should be further polished. Please consider the comments and suggestions from Report #2 carefully, and revise the manuscript further.

**Response:** Thank you for the encouragement. We revised the manuscript carefully, especially according to the comments from Report #2.

**A marked-up manuscript version**

[revised manuscript text omitted]

---

## Author Response (AR3)

**A point-by-point response to the comments by the editor**

Thank you for the helpful and constructive comments on our manuscript. Your comments and our responses are presented below.

1. This special issue focus on hazard, risk assessment, and management. More discussions should be focus these topics please. Please carefully discuss your research results around hazard and risk, and give suitable suggestions on management.

**Response:** We agree with this comment, and the additional data have been provided in the following sections: 1) Abstract, Page 1 Line 12-14, 31; 2) The first paragraph in the introduction, Page 2, Line 8, 11-14; 3) Study area, Page 6 Line 1; 4) 3.4.1 Distribution of SSC in four seasons, Page 12, Line 8-10, 19 and 23-24; 5)A paragraph was added to discuss the research results on hazards and risks and provide suitable suggestions on management in Page 13 Line 22-24 and Page 14 Line 1-7.

2. Some quantified results should be added to express your views better in abstract

**Response:** We agree with this comment, and quantified results were provided in the abstract, Page 1 Line 23-27.

3. Page 1 Line 30 Keyword Remote sensing inversion does not appear in the following text, please consider choosing a more suitable keyword.

**Response:** We agree with this comment, and the keyword "Remote sensing inversion" was rephrased as "Remote sensing".

4. Page 3 Lines 6-7 Please consider further supplementing the defects of the traditional vegetation indices.

**Response:** We agree with this comment. The defects of traditional vegetation indices are discussed in the third paragraph of the introduction, Page 3 Line 10-15.

5. Page 3 Lines 19-20 The reasons why the same inversion models cannot be applied to quantitatively analyze the SSC in different seasons should be supplemented

**Response:** We agree with this comment and included the reasons why the same inversion models cannot be applied to quantitatively analyze the SSC in different seasons in the fourth paragraph of the introduction, Page 3 Line 21-24 and Page 4 Line 1-6.

6. The shortcoming of the method and the uncertainty of the quantization should be mentioned.

**Response:** We agree with this comment and added the shortcoming of the method and the uncertainty of the quantization in the third paragraph of the discussion, Page 15 Line 8-12.

7. Language should be further polished before submission.

**Response:** We agree with this comment, and the language was further polished again.

A marked-up manuscript version

[revised manuscript text omitted]

---

## Author Response (AR4)

**A point-by-point response to the comments by the editor**

Thank you for your attention on our manuscript. The language was further
polished again, and the grammer errors and the format of references

5     were revised.

A marked-up manuscript version

[revised manuscript text omitted]